# Pathway-specific effects of ADSL deficiency on neurodevelopment

Ilaria Dutto[1†], Julian Gerhards[2,3†], Antonio Herrera[4], Olga Souckova[5], Václava Škopová[5], Jordann A Smak[6], Alexandra Junza[7,8], Oscar Yanes[7,8], Cedric Boeckx[9,10,11], Martin D Burkhalter[2], Marie Zikánová[5], Sebastian Pons[4], Melanie Philipp[2,3], Jens Lüders[1*], Travis H Stracker[1,6*]

[1]Institute for Research in Biomedicine, The Barcelona Institute of Science and Technology, Barcelona, Spain; [2]Department of Experimental and Clinical Pharmacology and Pharmacogenomics, University of Tübingen, Tübingen, Germany; [3]Institute of Biochemistry and Molecular Biology, Ulm University, Ulm, Germany; [4]Department of Cell Biology, Instituto de Biología Molecular de Barcelona, Barcelona, Spain; [5]Department of Paediatrics and Inherited Metabolic Disorders, First Faculty of Medicine, Charles University and General University Hospital in Prague, Prague, Czech Republic; [6]National Cancer Institute, Center for Cancer Research, Radiation Oncology Branch, Bethesda, United States; [7]Universitat Rovira i Virgili, Department of Electronic Engineering, IISPV, Tarragona, Spain; [8]CIBER de Diabetes y Enfermedades Metabólicas Asociadas (CIBERDEM), Instituto de Salud Carlos III, Madrid, Spain; [9]ICREA, Barcelona, Spain; [10]Institute of Complex Systems (UBICS), Universitat de Barcelona, Barcelona, Spain; [11]Section of General Linguistics, Universitat de Barcelona, Barcelona, Spain

*For correspondence:
jens.luders@irbbarcelona.org
(JL);
travis.stracker@nih.gov (THS)

†These authors contributed equally to this work

Competing interest: The authors declare that no competing interests exist.

**Abstract** Adenylosuccinate lyase (ADSL) functions in de novo purine synthesis (DNPS) and the purine nucleotide cycle. ADSL deficiency (ADSLD) causes numerous neurodevelopmental pathologies, including microcephaly and autism spectrum disorder. ADSLD patients have normal serum purine nucleotide levels but exhibit accumulation of dephosphorylated ADSL substrates, S-Ado, and SAICAr, the latter being implicated in neurotoxic effects through unknown mechanisms. We examined the phenotypic effects of ADSL depletion in human cells and their relation to phenotypic outcomes. Using specific interventions to compensate for reduced purine levels or modulate SAICAr accumulation, we found that diminished AMP levels resulted in increased DNA damage signaling and cell cycle delays, while primary ciliogenesis was impaired specifically by loss of ADSL or administration of SAICAr. ADSL-deficient chicken and zebrafish embryos displayed impaired neurogenesis and microcephaly. Neuroprogenitor attrition in zebrafish embryos was rescued by pharmacological inhibition of DNPS, but not increased nucleotide concentration. Zebrafish also displayed phenotypes commonly linked to ciliopathies. Our results suggest that both reduced purine levels and impaired DNPS contribute to neurodevelopmental pathology in ADSLD and that defective ciliogenesis may influence the ADSLD phenotypic spectrum.

## Editor's evaluation

This article attempts to elucidate adenylosuccinate lyase (ADSL) deficiency in neurodevelopment using a variety of approaches. It provides a better understanding of the underlying mechanisms that lead to microcephaly in some patients affected by ADSL deficiency.

## Introduction

Adenylosuccinate lyase (ADSL) is a conserved homotetrameric enzyme that catalyzes the eighth reaction in the de novo purine synthesis (DNPS) pathway and also participates in the purine nucleotide cycle (*Figure 1—figure supplement 1*; *Daignan-Fornier and Pinson, 2019*). Mutations in *ADSL* cause adenylosuccinate lyase deficiency (ADSLD), an autosomal recessive disorder characterized by defects in purine metabolism and heterogeneous neurological phenotypes that include lack of eye-to-eye contact, auto-aggressive behavior, speech impairment, mild psychomotor delay, transient contact defects, autism spectrum disorder, epilepsy, and in some cases, microcephaly, encephalopathy, ataxia, or coma vigil (*Jaeken and Van den Berghe, 1984*; *Jurecka et al., 2015*). While the incidence of ADSLD has not been fully established, over 100 patients have been diagnosed to date and subcategorized based on their symptoms that range from premature death to milder developmental and behavioral disorders (*Jurecka et al., 2015*; http://www.adenylosuccinatelyasedeficiency.com/).

ADSLD can be diagnosed by detecting elevated levels of the substrates of ADSL, SAICAR (also referred to as SZMP), and S-AMP, as well as their dephosphorylated riboside forms, SAICAr and S-Ado, in bodily fluids (*Jaeken and Van den Berghe, 1984*). As normal levels of purine nucleotides were detected in serum from ADSLD patients, the accumulation of S-Ado, and particularly SAICAr, was proposed to play a role in the disease pathology (*Jaeken and Van den Berghe, 1984*; *Jurecka et al., 2015*; *Stone et al., 1998*).

In yeast, ADSL (Ade13) loss provokes genomic instability and is lethal (*Daignan-Fornier and Pinson, 2019*; *Giaever et al., 2002*; *Pinson et al., 2009*; *Chen et al., 2016*). Lethality in yeast can be rescued by deletion of a number of DNPS enzymes upstream of ADSL, or the transcription factors that regulate the pathway, indicating that the accumulation of metabolic intermediates, rather than impaired DNPS, underlies the toxicity (*Daignan-Fornier and Pinson, 2019*; *Chen et al., 2016*). In *Caenorhabditis elegans*, ADSL loss caused delayed growth, infertility, reduced life span, and locomotion defects. In some studies, growth, life span, and locomotion could be linked to the accumulation of SAICAR (*Chen et al., 2016*; *Marsac et al., 2019*; *Fenton et al., 2017*).

Perfusion of rat brains with SAICAr caused cellular attrition in the hippocampus, leading to the proposition that SAICAr accumulation is neurotoxic, although the potential mechanism remains unknown (*Stone et al., 1998*). In glucose-deprived cancer cells, SAICAR accumulation was shown to activate PKM2, and a number of other kinases, to promote cancer survival in glucose-limiting conditions, suggesting that purine metabolite accumulation could have distinct signaling outcomes that impact on cell behavior and fate during development (*Keller et al., 2012*; *Keller et al., 2014*; *Yan et al., 2016*). However, despite extensive enzymology and structural information, the underlying mechanisms by which neuropathology arises in ADSLD remain unknown.

To address the potential roles of ADSL deficiency in neurodevelopment, we systematically examined the consequences of ADSL depletion in diploid human cells and in vivo. We found that ADSL depletion in human epithelial cells impaired cell cycle progression, induced DNA damage signaling, and impaired primary ciliogenesis. Deletion of p53 or supplementation with nucleosides rescued cell cycle progression and DNA damage signaling, respectively. In contrast, ciliogenesis defects were unaffected by p53 status or nucleoside supplementation and could be reproduced by SAICAr supplementation or attenuated by the inhibition of phosphoribosylaminoimidazole carboxylase (PAICS), which is required to generate SAICAR/r upstream of ADSL. Depletion of ADSL in chicken or zebrafish embryos impaired neurogenesis and caused developmental defects. Both chicken and zebrafish embryos exhibited microcephaly, which is observed in some ADSLD patients, and increased DNA damage was evident in zebrafish. In addition, fish embryos displayed ciliopathy-related phenotypes and interventions to limit SAICAR accumulation-rescued impaired neurogenesis. Together, our results indicate that ADSL depletion causes context-dependent phenotypes associated with its role in both the purine nucleotide cycle and DNPS, which together impact neurodevelopment.

## Results

### ADSL depletion causes p53-dependent proliferation defects

To investigate the impact of ADSL on cellular homeostasis, we depleted ADSL with a pool of four siRNAs in hTERT-immortalized human retinal epithelial cells (hTERT-RPE-1, referred to henceforth as RPE-1). Depletion of ADSL was effective as we observed 80% depletion of the mRNA and a clear

reduction in protein levels (*Figure 1A and B*). This was accompanied by reduced levels of AMP and GMP, as well as accumulation of S-Ado (*Figure 1C*, *Figure 1—figure supplement 2A and B*). SAICAr levels were at the limit of detection in RPE-1 cells and differences could not be assessed (*Figure 1— figure supplement 2B*), but elevated SAICAr could be readily observed following ADSL depletion in HeLa, potentially due to elevated levels of DNPS enzymes that have been observed in cancer cells (*Figure 1—figure supplement 2C*; *Zurlo et al., 2019*; *Agarwal et al., 2020*; *Yamauchi et al., 2022*; *Taha-Mehlitz et al., 2021*). We deconvolved the siRNA pool and identified an effective single siRNA (#2) to further validate results from the siRNA pool (*Figure 1—figure supplement 2D*).

As ADSL is critical for DNPS, we examined cell growth following ADSL depletion and found reduced levels of proliferation in ADSL-depleted cells compared to controls (*Figure 1D*). ADSL-depleted cells frequently lacked Ki67 expression, indicating that some cells were exiting the cell cycle, had increased levels of p53, and showed accumulation in G1 phase of the cell cycle (*Figure 1E and F*, *Figure 1— figure supplement 2E*). Trypan Blue and β-galactosidase assays indicated that cell death was not increased (*Figure 1—figure supplement 2F*) and that the Ki67-negative cells were likely quiescent and not senescent (*Figure 1—figure supplement 2G*). Deletion of *TP53* rescued proliferation, prevented G1 arrest, and restored the number of Ki67-positive cells (*Figure 1G*, *Figure 1—figure supplement 2H and I*). The reduction in Ki67-positive cells could be rescued by stable expression of an siRNA-resistant allele of *ADSL* (ADSL*) (*Figure 1H*, *Figure 1—figure supplement 2J*). We also checked whether ADSL-depleted RPE-1 cells underwent differentiation by staining with vimentin, a marker of undifferentiated cells (*Tamiya et al., 2010*), and cytokeratin 20 (CK20), a marker of differentiation, upon ADSL depletion. We did not observe any CK20 signal or a reduction in vimentin-positive cells in the population upon ADSL silencing compared to the controls, arguing against premature differentiation (*Figure 1—figure supplement 2K*).

To identify the cause of cell cycle exit in ADSL-depleted cells, we performed specific treatments. To restore purine levels, we supplemented cells with a mixture of nucleosides or with adenosine alone. To reduce potential accumulation of SAICAr that is generated by DNPS, we treated with MRT00252040, a small-molecule inhibitor of PAICS, the enzyme required to generate SAICAr upstream of ADSL (*Figure 1—figure supplement 1*; *Hoxhaj et al., 2017*). Supplementation of ADSL-depleted RPE-1 cells with nucleosides or adenosine did not prevent p53 induction or cell cycle exit (*Figure 1I and J*). Similarly, treatment with MRT00252040 did not influence p53 induction or loss of Ki67 (*Figure 1K and L*). This demonstrated that ADSL depletion in non-transformed human epithelial cells leads to a partial p53-dependent cell cycle exit/arrest that is not rescued by complementing the defects of ADSL depletion with nucleoside supplementation or inhibiting PAICS to prevent SAICAR/r production upstream of ADSL.

## ADSL depletion causes elevated DNA damage signaling

Reduced levels of purine nucleotides in ADSL-depleted cells may cause replication stress and DNA damage (*Kim et al., 2015*; *Benedict et al., 2018*; *Gottifredi et al., 2001*). To address this, we examined the levels of chromatin-bound RPA, a surrogate marker of ssDNA accumulation and indicator of replication stress. In ADSL-depleted cells, chromatin-bound RPA levels were increased, indicative of replication stress, although the effect was mild (*Figure 2A*). Additionally, we observed an increased number of cells with more than five 53BP1 foci per cell, indicative of DNA double-strand break (DSBs) accumulation (*Figure 2B*). 53BP1 foci were reduced by treatment with a small-molecule inhibitor for ATM (*Figure 2C*), indicating an active DNA damage response. Supplementation of cells with nucleosides suppressed the appearance of DNA DSBs detected by 53BP1 following ADSL depletion or treatment with the ribonucleotide reductase inhibitor hydroxyurea (HU) that depletes dNTP pools (*Figure 2D and E*). A similar reduction in 53BP1 foci was also observed with adenosine supplementation of ADSL-depleted cells (*Figure 2F*). Nucleoside supplementation also reduced the increase in γH2AX, the phosphorylated form of the histone variant H2AX, which is also a marker of DSBs (*Figure 2G*). In contrast to nucleoside supplementation, the PAICS inhibitor MRT00252040 did not rescue the increased levels of DSBs (*Figure 2H*). These data indicate that ADSL depletion in cultured cells induces mild levels of DNA damage signaling that can be suppressed by nucleoside supplementation, but not by the inhibition of PAICS. This implicates defects in the purine nucleotide cycle, rather than impaired DNPS or specific metabolite accumulation resulting from ADSL depletion, in replication stress and DSB formation. In addition, it indicates that the partial p53-dependent cell cycle exit is

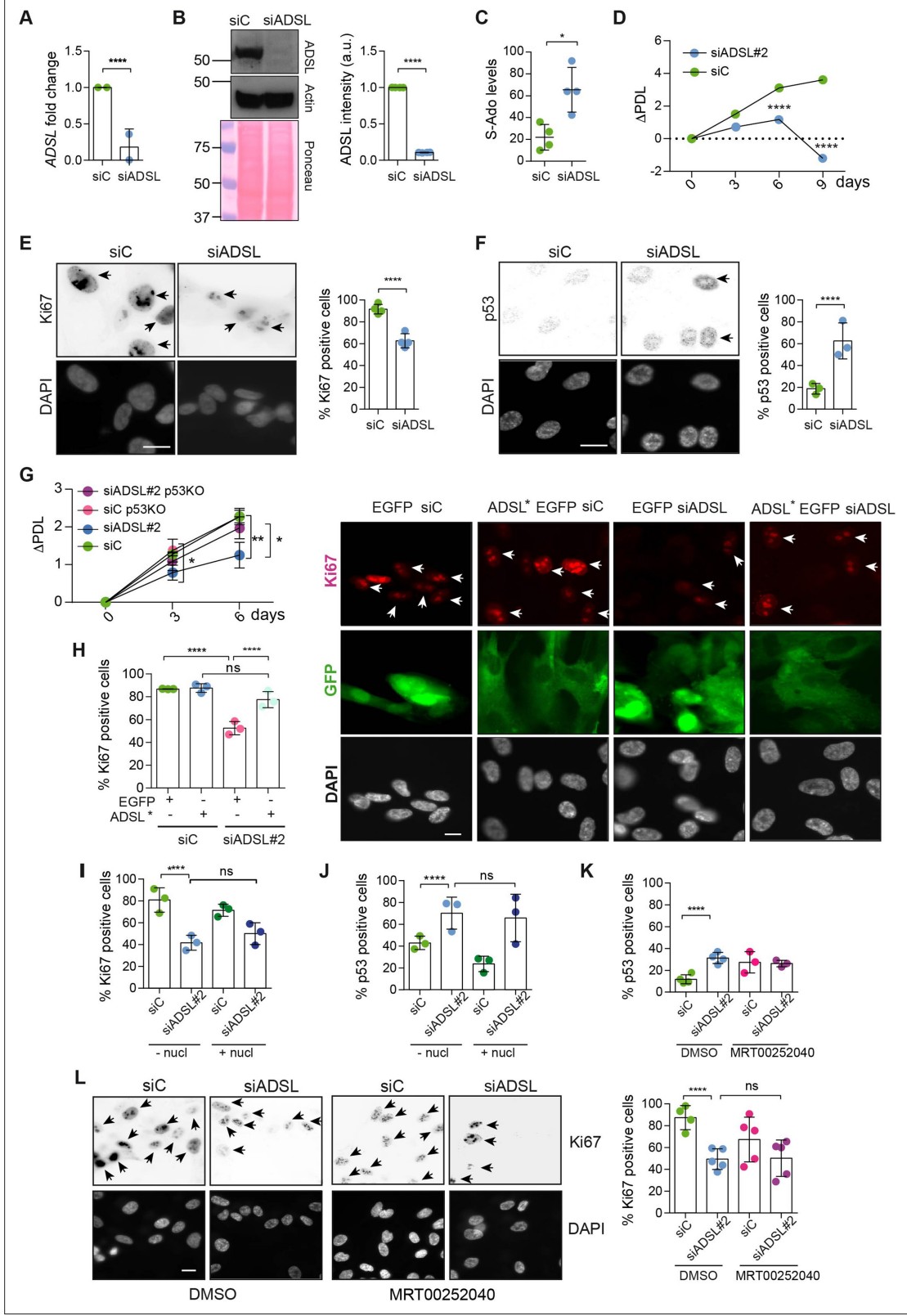

**Figure 1.** Adenylosuccinate lyase (ADSL) depletion causes p53-dependent proliferation defects. (**A**) Reduced mRNA levels of *ADSL* confirmed by qRT-PCR experiments. hTERT-RPE-1 were silenced with smart pool RNAi for 96 hr before harvesting. Two independent experiments in triplicate are shown in the panel (n = 2, two-tailed *t*-test, ****p<0.0001) (**B**) Western blot of RPE-1 cell extracts treated as in (**A**). One experiment is shown as representative of four independent experiments. Actin was used as a loading control. Quantifications of ADSL intensity in four different experiments were performed by

*Figure 1 continued on next page*

*Figure 1 continued*

ImageJ software and normalized to actin first and then to the relative controls (n = 4, two-tailed *t*-test, ****p<0.0001). Ponceau is shown as an additional loading and transfer control. (**C**) S-Ado levels in RPE-1 cells silenced with a single control or ADSL siRNA (n = 4, two-tailed *t*-test, *p<0.05). (**D**) Cell proliferation rates of RPE-1 cells quantified every 3 days after treatment with a single control or *ADSL* siRNA in medium with serum (n = 3, two-tailed *t*-test ****p<0.0001). ΔPDL represent the difference in population doubling levels quantified through the formula described in Materials and methods. (**E**) Ki67-positive cells (indicated by arrows) upon 96 hr of silencing with control or *ADSL* smart pool siRNAs. Scale bar 10 µm (n = 4, scored 767 cells for siC, 694 cells for siADSL conditions, ****p<0.0001). (**F**) The percentage of p53-positive cells (indicated by arrows) following treatment with control or ADSL smart pool siRNAs was quantified in three independent experiments (n = 3, scored 835 cells for siC and 1126 cells for siADSL, *p<0.05). (**G**) Cell proliferation rates in RPE-1 wt and p53 knockout KO cells as in (**D**) were counted for 6 days (n = 3, two-tailed *t*-test, **p<0.01, *p<0.05). (**H**) RPE-1 cells stably expressing EGFP or ADSL*-EGFP were transfected with a single control or ADSL siRNAs for 96 hr and immunostained with anti-Ki67 antibody. Scale bar = 20 µm. Quantification of Ki67-positive cells (n = 3, scored 278 cells for EGFP siC, 264 for EGFP siADSL, 266 for ADSL* siC, 232 cells for ADSL* siADSL conditions, *ns* not significant, ****p<0.0001). (**I**) Quantification of RPE-1 transfected with a single control or ADSL siRNA for 96 hr in the presence or absence of 60 µM nucleosides. Cells were fixed and immunostained with anti-Ki67 antibody (n = 3, at least 100 cells per conditions were counted in each experiment, *ns* not significant, ****p<0.0001). (**J**) Quantification of RPE-1 in the same conditions of (**I**) and immunostained with anti-p53 antibody (n = 3, at least 100 cells were counted for each condition per experiment, *ns* not significant, ****p<0.0001). (**K**) Quantification of p53-positive cells in *ADSL*-depleted cells in the presence or absence of the phosphoribosylaminoimidazole carboxylase (PAICS) inhibitor MRT00252040 (n = 3, scored 223 cells for siC, 248 cells for siADSL, 336 cells for siC+ MRT00252040, 365 cells for siADSL + MRT00252040, *ns* not significant, ****p<0.001, *p<0.05). (**L**) Quantification of Ki67-positive cells in ADSL-depleted cells in the presence or absence of MRT00252040 (n = 5, more than 60 cells were counted in each condition for each experiment, *ns* not significant, ***p<0.001). Positive cells are indicated with arrows in siADSL panels. All graphs depict means ± SD with individual values shown in circles.

The online version of this article includes the following source data and figure supplement(s) for figure 1:

**Source data 1.** Related to *Figure 1A and C–L*.

**Source data 2.** Related to *Figure 1B*.

**Figure supplement 1.** The role of adenylosuccinate lyase (ADSL) in de novo purine synthesis (DNPS) and the purine nucleotide cycle.

**Figure supplement 2.** Adenylosuccinate lyase (ADSL) depletion does not cause senescence or promote differentiation.

**Figure supplement 2—source data 1.** Related to *Figure 1—figure supplement 2*.

not solely a consequence of DNA damage signaling or purine metabolite accumulation, as it was not rescued by either intervention.

## ADSL depletion impairs neurogenesis in the developing chicken neural tube

Given the effects of ADSL depletion on cell growth and proliferation, we sought to examine the consequences of its loss in vivo. To this end, we used the chicken embryo system to examine the influence of ADSL depletion on nervous system development. We electroporated one side of the neural tube with plasmid expressing *GFP* as a transfection marker in combination with either control or ADSL shRNA vectors. After confirming efficient ADSL depletion (*Figure 3A*), we evaluated neurogenesis by staining with markers for proliferating neural progenitors (SOX2 positive) and post-mitotic neurons (ELAVL3/4 positive). We found that in the ADSL-depleted side both cell populations were reduced when compared to the non-transfected side (*Figure 3B*) and that the size of the tissue was smaller, suggesting reduced growth and/or increased cell death. Staining for the apoptotic marker cleaved caspase-3 (CC3) revealed no notable differences, suggesting that this was not due to increased cell death (*Figure 3—figure supplement 1A*).

We then analyzed SOX2 and ELAVL3/4 staining only within the GFP-positive transfected cells and found that ADSL depletion increased the percentage of SOX2-positive progenitors relative to ELAVL3/4-positive neurons (*Figure 3C*, *Figure 3—figure supplement 1B*). This suggested that reduced tissue growth was not due to premature differentiation but possibly due to a proliferation defect in the progenitor population. To study cell cycle progression in neural stem cells, we performed fluorescence-associated cell sorting (FACS) analysis of GFP-positive, ELAVL3/4-negative cells following electroporation of control or ADSL shRNA. We found that there was a slight increase in the G2/M population after ADSL depletion (*Figure 3D*). Further analysis of stained tissue sections showed that ADSL depletion caused a reduction in the fraction of cells that incorporated EdU and an increase in the fraction of cells positive for the G2/M marker phosphorylated histone H3-Ser10 (pH3S10) (*Figure 3E*). We separated the pH3S10-positive cells into two populations: G2 cells, identified by punctate pH3S10 staining, and mitotic cells, displaying broadly distributed pH3S10 staining.

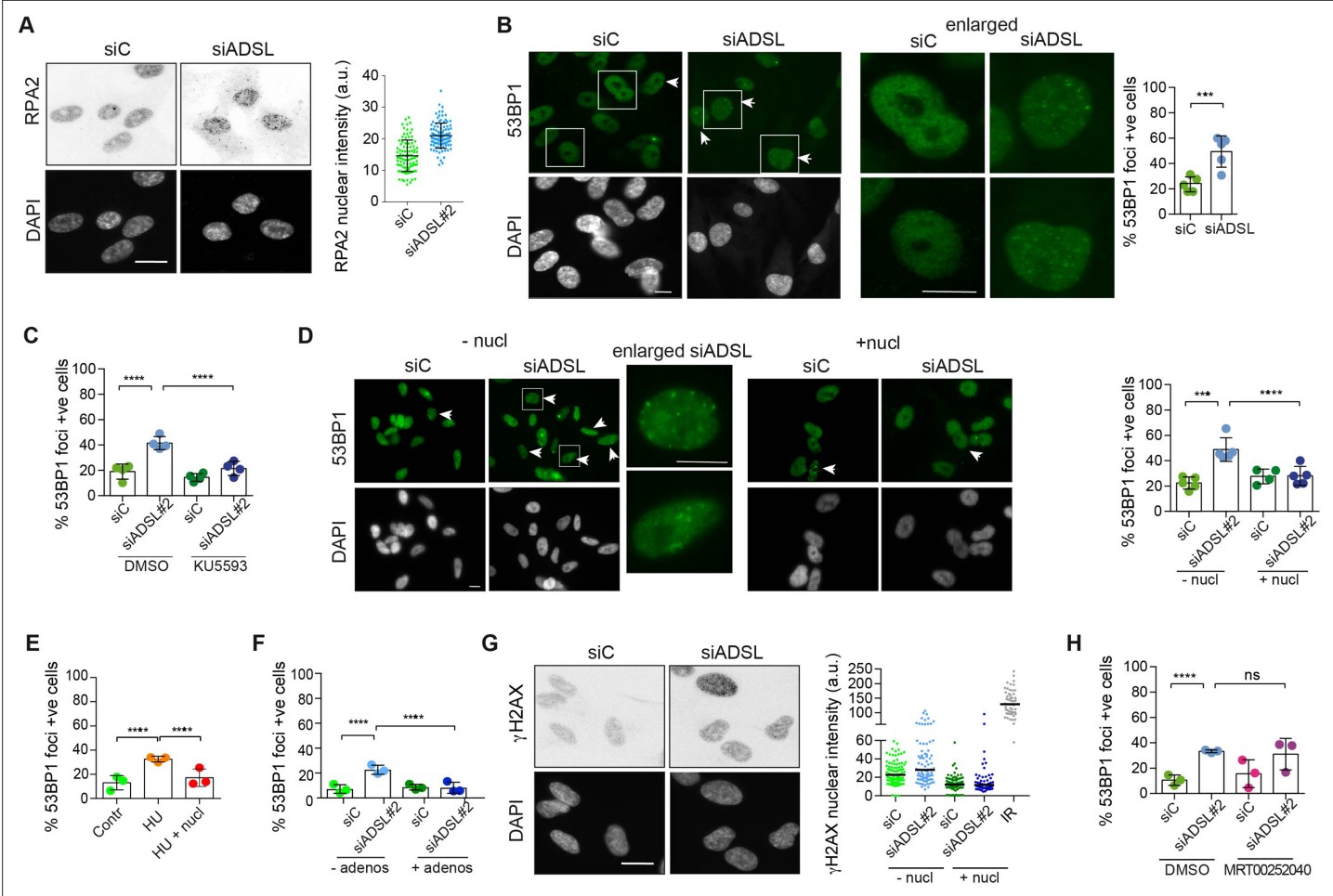

**Figure 2.** Adenylosuccinate lyase (ADSL) depletion caused elevated DNA damage signaling. (**A**) RPE-1 were silenced for 96 hr with a smart pool of ADSL siRNAs, chromatin extracted, and immunostained with an anti-RPA2 antibody. Nuclear intensity of cells was measured- Quantification of one representative experiment of four that showed similar results is shown; SD and average are indicated in black. After normalization to the average of the control (siC), two-tailed *t*-test was used for statistical analysis of n = 4 independent experiments: *p<0.05 was observed for siADSL to siC. (**B**) RPE-1 were silenced for 96 hr with a smart pool of ADSL siRNAs, fixed, and immunostained with anti-53BP1 antibody. Scale bar = 10 μm. Quantification of positive cells that have more than five foci per cell (n = 5 scored 1134 cells for siC, 1144 cells for siADSL, ***p<0.001). (**C**) RPE-1 were silenced with a single control or ADSL siRNA with or without 5 mM ATM inhibitor (KU5593) (n = 4, scored 359 cells for siC, 322 cells for siADSL, 307 cells for siC + KU5593, 279 cells for siADSL + KU5593, ****p<0.0001). (**D**) Cells were silenced for 96 hr, treated or not with 60 μM (1×) nucleosides, and stained for 53BP1. Scale bar = 10 μm (n = 5, scored 717 cells for siC, 608 cells for siADSL, 576 cells for siC + nucl, 512 cells for siADSL + nucl, ****p<0.0001). (**E**) RPE-1 cells were treated with 2 mM hydroxyurea (HU) for 6 hr and treated or not with 1× nucleosides and stained for 53BP1. Cells with more than five foci per nucleus were counted as positive. Three independent experiments were performed (n = 3, at least 100 cells per experiment per condition were counted, ****p<0.0001). (**F**) Cells were silenced for 96 hr, treated or not with 80 μg/ml adenosine, and stained for 53BP1 (n = 3, scored 544 cells for siC, 428 cells for siADSL, 485 cells for siC+ adenosine, 411 cells for siADSL + adenosine, ****p<0.0001). (**G**) RPE-1 treated as in (**A**) were fixed and stained for γH2AX (H2AX phosphorylated on Ser-139). Scale bar = 10 μm. 5 Gy X-ray irradiation (IR) was used as positive control. Quantification of one representative experiment of two that showed similar results is shown; median is indicated in black. After normalization to the average of the control (siC), one-tailed *t*-test was used for statistical analysis of n = 3 independent experiments: *p<0.05 was observed for siADSL (to siC), and for siADSL relative to siADSL + nucl. There is no statistical difference between siC and siC + nucl. (**H**) RPE-1 were silenced in the presence or absence of 4 μM MRT00252040, fixed and stained for 53BP1 (n = 4, scored 367 cells for siC, 313 cells for siADSL, 294 cells for siC + MRT00252040, 241 cells for siADSL + MRT00252040, *ns* not significant, ****p<0.0001). All bar graphs show means ± SD with individual values in circles.

The online version of this article includes the following source data for figure 2:

**Source data 1.** Related to *Figure 2*.

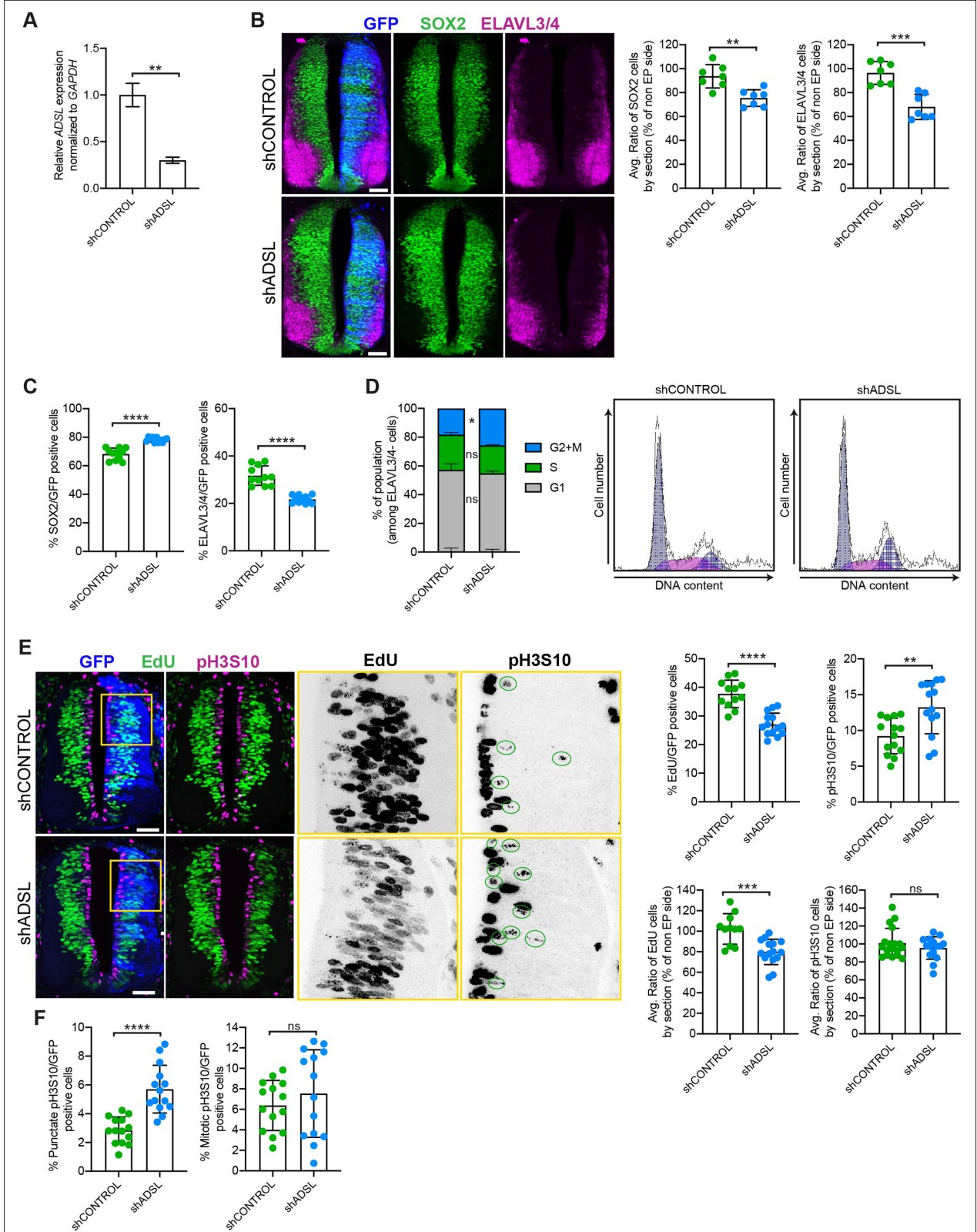

**Figure 3.** Adenylosuccinate lyase (ADSL) depletion causes neurodevelopmental delay in the chicken neural tube. (**A**) mRNA levels of *ADSL* and *GAPDH* were measured by qRT-PCR in chicken embryonic fibroblasts (CEFs) transfected for 24 hr with shCONTROL or shADSL to confirm knockdown efficiency (n = 3, two-tailed *t*-test, \*\*p<0.01). (**B**) Transverse sections of HH12 chicken neural tubes 48 hr post electroporation (hpe) with shCONTROL or shADSL plasmids and stained with antibodies against SOX2 (green) and ELAVL3/4 (magenta). Transfection was detected by GFP (blue). Scale bar

*Figure 3 continued on next page*

*Figure 3 continued*

= 50 μm. Average ratio of neural stem cells (NSCs, SOX2+) 48 hpe with shCONTROL or shADSL obtained by comparing the mean number of SOX2+ cells on the electroporated and non-electroporated side (n = 7 embryos, two-tailed *t*-test, **p<0.01). Average ratio of cells differentiated into neurons (ELAVL3/4) at 48 hpe with shCONTROL or shADSL obtained by comparing the mean number of ELAVL3/4-positive cells on the electroporated and the non-electroporated side (n = 7 embryos, two-tailed *t*-test, ***p<0.001). (**C**) Percentage of electroporated cells identified as NSCs (SOX2) or neurons (ELAVL3/4) 48 hpe with shCONTROL or shADSL (n = 11 embryos, two-tailed *t*-test, ****p<0.0001). (**D**) The cell cycle profiles of NSCs (GFP+/ELAVL3/4-) obtained by fluorescence-associated cell sorting (FACS) 48 hpe with shCONTROL or shADSL into HH12 chicken neural tubes. The mean of two independent experiments is shown in the left panel. 6–8 embryos per condition were used for each experiment. Two-tailed *t*-test was used for statistical analysis of n = 2 independent experiments, *ns* not significant, *p<0.05. Cell cycle profiles of a representative experiment are shown in the right panels. (**E**) Transverse sections of HH12 chicken neural tubes 48 hpe with shCONTROL or shADSL plasmids, and stained with EdU (green) and an antibody against pH3S10 (magenta). Transfection was detected by GFP (blue). Scale bar = 50 μm. Areas indicated in yellow are amplified in the right panels showing separated channels in black. Green circles in pH3S10 amplification show punctate pH3S10-positive cells. Percentage of transfected cells identified as EdU-positive 48 hpe with shCONTROL or shADSL (n = 12 embryos [shCONTROL] and 14 embryos [shADSL], two-tailed *t*-test, ****p<0.0001). Percentage of pH3S10 among the GFP+ cell population 48 hpe with shCONTROL or shADSL (n = 14 embryos, two-tailed *t*-test, *ns* not significant, **p<0.01, ****p<0.0001). Average ratio of EdU and pH3S10-positive cells 48 hpe of shCONTROL or shADSL plasmids, obtained by comparing the mean number of EdU cells on the electroporated and the non-electroporated side (EdU: n = 11 embryos [shCONTROL], 15 embryos [shADSL], two-tailed *t*-test, ***p<0.001; pH3S10: n = 18 embryos [shCONTROL], 15 embryos [shADSL], two-tailed *t*-test, *ns* not significant). (**F**) Percentage of punctate pH3S10 (G2 phase) and mitotic pH3S10 (M phase) among the GFP+ cell population 48 hpe of shCONTROL or shADSL plasmids (n = 14 embryos, two-tailed *t*-test, *ns* not significant, ****p<0.0001). Bar graphs show means ± SD.

The online version of this article includes the following source data and figure supplement(s) for figure 3:

**Source data 1.** Related to *Figure 3*.

**Figure supplement 1.** Lack of cell death or increased differentiation in developing adenylosuccinate lyase (ADSL)-depleted chicken neural tubes.

This revealed that only the G2 fraction of cells was increased by ADSL depletion, indicating that ADSL depletion caused a specific delay in G2 phase in the SOX2+ population, rather than during mitosis (*Figure 3F*). Together, our in vitro data indicate that ADSL depletion leads to a mild induction of DNA damage signaling and impaired cell cycle progression. In vivo, this manifests as reduced cellularity in the developing brain, without a clear induction of cell death or senescence.

## Ciliogenesis defects following ADSL depletion

As there are non-cycling cells in the brain and ADSL depletion caused cell cycle exit in RPE-1 cells, a condition frequently accompanied by ciliogenesis, we tested the ability of control and ADSL-depleted RPE-1 cells to assemble cilia. Following treatment with siRNA, cells were serum-starved for 48 hr and analyzed by immunofluorescence microscopy. Ki67 staining confirmed that most of the cells in both conditions exited the cell cycle (*Figure 4A*). We next examined ciliogenesis by staining for the ciliary marker ARL13B and the centrosome marker pericentrin (PCNT). Fewer cells treated with the ADSL siRNA pool had cilia, and the cilia that were present were shorter when compared to controls (*Figure 4B*). We also observed shorter cilia upon depletion with single siRNAs for ADSL (*Figure 4—figure supplement 1A*). To exclude the possibility that ciliogenesis was simply delayed, we quantified the number of ciliated cells 72 hr after serum starvation and observed a similar defect (*Figure 4—figure supplement 1B*). Defective ciliogenesis was rescued by expression of an siRNA-resistant cDNA (ADSL*), but not by nucleoside supplementation (*Figure 4C and D*). Inhibition of the DNPS pathway with methotrexate (MTX), which impairs steps in DNPS up- and downstream of ADSL (*Figure 1—figure supplement 1*), had no effect on ciliogenesis in control cells (*Figure 4—figure supplement 1C*), but rescued both the number of ciliated cells and cilia length in ADSL-depleted RPE-1 cells (*Figure 4—figure supplement 1D*).

Since defective ciliogenesis caused by ADSL depletion was rescued by MTX, but not by nucleoside supplementation, we next examined whether specific inhibition of PAICS was sufficient to induce the phenotype. Following ADSL depletion, we treated cells with the PAICS inhibitor MRT00252040 (*Hoxhaj et al., 2017*). Similar to MTX, this rescued ciliogenesis as the number and length of cilia were similar in control and ADSL-depleted cells (*Figure 4E*). As inhibition of PAICS impairs DNPS upstream of SAICAR production, we treated cells with SAICAr and observed that this recapitulated the ciliogenesis defect observed in ADSL-depleted cells (*Figure 4F and G*). SAICAr treatment did not activate AKT activity, which has recently been shown to inhibit ciliogenesis due to serum starvation (*Figure 4—figure supplement 1E*; *Walia et al., 2019*). To exclude additional indirect effects as a cause for the reduction in the number of ciliated cells, such as cell cycle progression defects caused

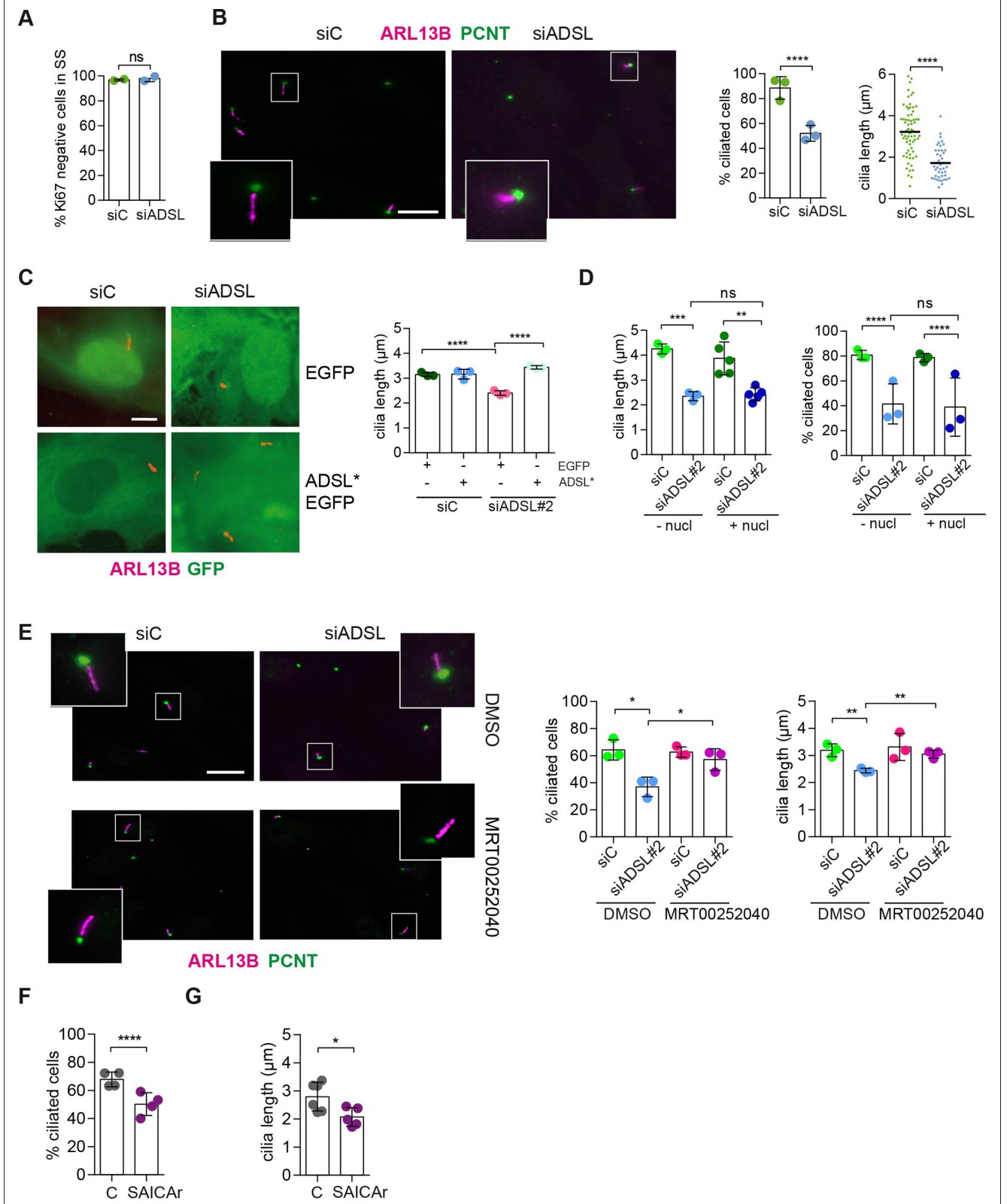

**Figure 4.** SAICAr-dependent ciliogenesis defects following adenylosuccinate lyase (ADSL) depletion. (**A**) RPE-1 were transfected with control or ADSL smart pool siRNAs. After 96 hr, cells were serum starved for 48 hr to induce ciliogenesis followed by staining against Ki67 and negative cells quantified (n = 2, two-tailed *t*-test, *ns* not significant). (**B**) Ciliated cells silenced as in (**A**) were stained for ARL13B (magenta) and pericentrin (PCNT) (green). Scale bar = 10 μm. Magenta squares show enlargements of the areas. Graphs show quantification of ciliated cells and cilia length (line indicates median)

*Figure 4 continued on next page*

*Figure 4 continued*

(n = 3, scored 108 cells for siC, 82 cells for siADSL, ****p<0.0001). (**C**) RPE1 cells stably expressing EGFP or ADSL*-EGFP were silenced for 96 hr with control or a single ADSL siRNA, serum starved for 48 hr, fixed, and stained for ARL13B (red). Scale bar = 5 μm. Graphs summarize three experiments (one-way ANOVA, *ns* not significant, ***p<0.001, **p<0.01, *p<0.05). (**D**) RPE-1 cells silenced with a single *ADSL* siRNA (siADSL#2) for 96 hr in the absence or presence of 1× nucleosides. Cilia frequency and cilia length were quantified; mean ± SD of n = 3 siC (scored 557 cells) and siADSL (scored 499 cells), n = 5 for siC (588 cells) and siADSL (scored 297 cells) with nucleosides, *ns* not significant, ***p<0.001. (**E**) RPE-1 cells were ADSL-depleted, treated or not with MRT00252040 and serum starved, and then immunostained for ARL13B (magenta) and PCNT (green). Cilia frequency and cilia length were quantified (n = 3, scored 261 cells for siC, 166 cells for siADSL, 287 cells for siC + MRT0025204, 170 cells for siADSL + MRT00252040, **p<0.01, ****p<0.0001). Scale bar = 10 μm. (**F**) Quantification of the cilia frequency in control and SAICAR-treated cells (n = 4, scored 589 cells for control, 456 cells for SAICAR-treated, ****p<0.0001). (**G**) Cilia length measurement of cells treated as in (**F**) (n = 5, two-tailed *t*-test, *p<0.05).

The online version of this article includes the following source data and figure supplement(s) for figure 4:

**Source data 1.** Related to *Figure 4* and *Figure 4—figure supplement 1*.

**Figure supplement 1.** Adenylosuccinate lyase (ADSL) depletion impairs ciliogenesis that can be rescued by methotrexate (MTX) treatment.

**Figure supplement 1—source data 1.** *Figure 4—figure supplement 1E*.

by DNA damage, p53 activation or defective cilia removal from interphase cells, we repeated the experiment in p53 KO cells and examined cilia resorption following the addition of serum. While the overall percentage of ciliated cells was slightly lower in p53 KO cells, depletion of ADSL recapitulated the result obtained in RPE-1 wt cells, a reduction in ciliated cells and cilia length compared to controls (*Figure 4—figure supplement 1F*). In addition, the kinetics of cilia removal following serum add back was similar in both controls and ADSL-depleted cells (*Figure 4—figure supplement 1G*). We concluded that impairing DNPS specifically at the ADSL-dependent step and/or SAICAr accumulation caused by ADSL depletion impairs the generation of primary cilia.

## ADSL depletion and SAICAr accumulation impair CP110 removal

To understand the origin of the ciliogenesis defect, we examined centriole configurations since mother centrioles, after conversion to basal bodies, template formation of the primary cilium. Centrosomes in ADSL-depleted cells had normal levels of PCNT and a normal number of centrioles (*Figure 5A*, *Figure 5—figure supplement 1A*). However, we found that the removal of CP110 from the mother centriole, a key step in early ciliogenesis, was impaired in serum-starved, ADSL-depleted cells. Compared to controls, a larger number of ADSL-depleted cells contained centrosomes with two CP110 foci (*Figure 5B*). This could be phenocopied by administration of SAICAr and was rescued by PAICS inhibition (*Figure 5C and D*). To determine if the retention of CP110 could underlie the phenotype, we co-depleted CP110 with ADSL using three different siRNAs. All three siRNAs silenced CP110, as verified by Western blot (*Figure 5—figure supplement 1B*) and partially depleted CP110 at centrioles (*Figure 5—figure supplement 1C*). In non-serum-starved conditions, CP110 siRNA-treated cells had fewer than the two centriolar CP110 foci typically observed in control cells (*Figure 5—figure supplement 1D*). The remaining centriolar signal was associated with daughter centrioles (distal to the base of the cilium in ciliated cells; *Figure 5—figure supplement 1C*). Co-depletion of CP110, using three independent siRNAs, with ADSL rescued the ciliogenesis defect (*Figure 5E*). These data demonstrated that ADSL deficiency or SAICAr administration impairs primary ciliogenesis, and this can be rescued by CP110 depletion or inhibition of PAICS, but not by supplementation of purine levels.

## Depletion of Adsl in zebrafish results in developmental defects

To test whether ADSL deficiency caused ciliary defects in vivo, we employed a zebrafish model. As CRISPR/Cas9-mediated gene knockout did not yield viable mutants, we used two different anti-sense morpholino oligonucleotides (MO) to deplete Adsl in zebrafish embryos. *Adsl* is ubiquitously expressed at early embryonic stages and, by the 18-somite stage, highly expressed in several areas of the developing brain, including the midbrain and mesencephalon (*Figure 6—figure supplement 1A–L*). Antibody staining demonstrated expression of Adsl in neurons, which was abolished upon injection of either MO (*Figure 6—figure supplement 2*). Examination of embryo morphology 48 hr post fertilization (hpf) revealed pericardial edema, kinked tail, hydrocephalus, and pinhead (microcephaly) phenotypes (*Figure 6A–E*). Defects in head size, which are consistent with the clinical presentation of ADSLD patients, were further corroborated by staining for skull formation that is coordinated

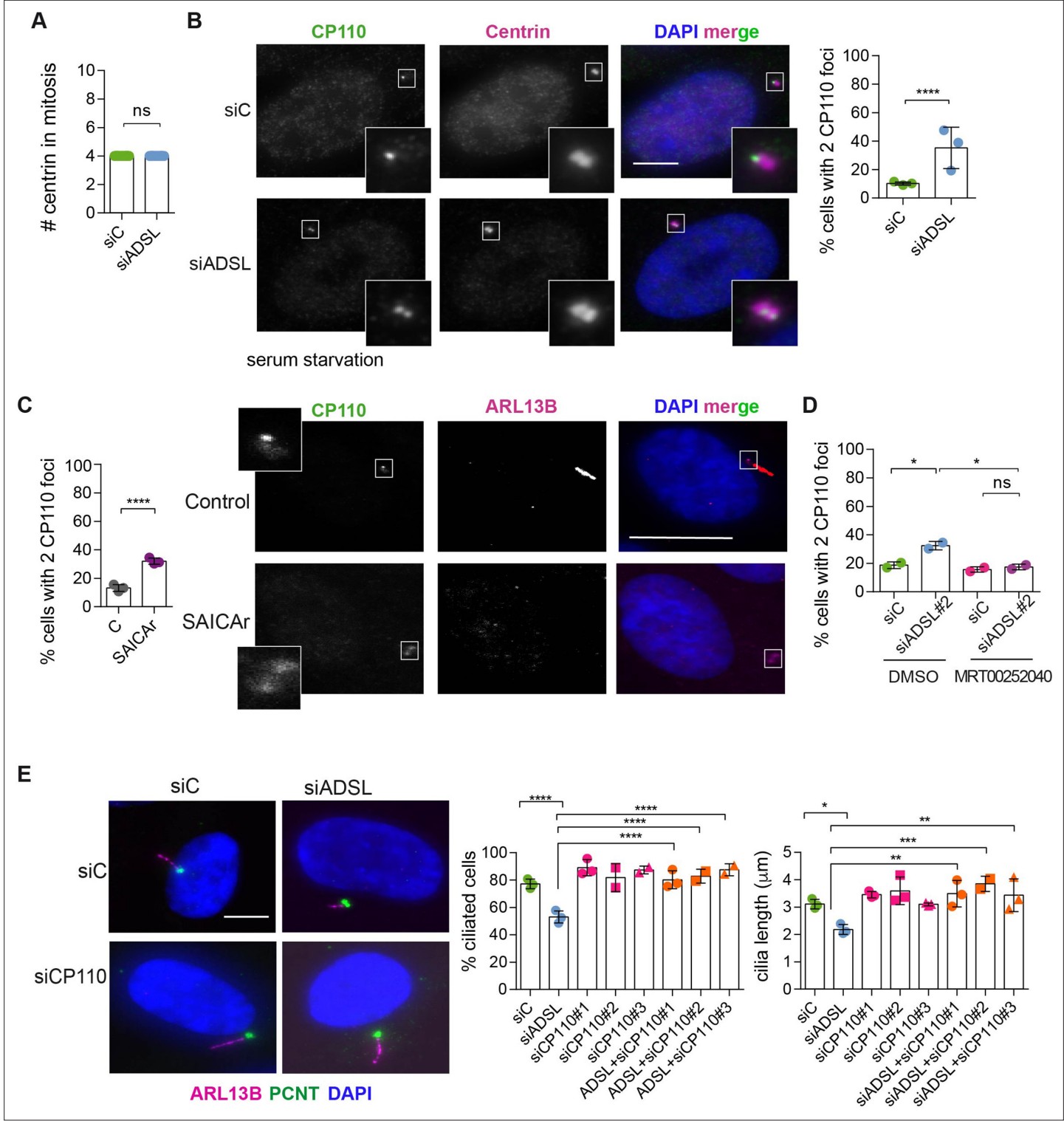

**Figure 5.** Adenylosuccinate lyase (ADSL) depletion and SAICAr impair CP110 removal. (**A**) Quantification of the number of centrin foci present in mitotic RPE-1 cells transfected with control or ADSL smart pool siRNAs for 96 hr (n = 2, two-tailed *t*-test, *ns* not significant). (**B**) ADSL-silenced cells and controls were stained for centrin (magenta) and CP110 (green). Nuclei are shown by DAPI (blue). Graph depicts the number of ciliated cells with two CP110 foci per centrosome (n = 3, scored 342 cells for siC, 221 cells for siADSL, *p<0.05). (**C**) Cells mock or treated with SAICAr were processed and analyzed as described in panel (**B**) (n = 3, scored 288 cells for control and 253 cells for SAICAr, ***p<0.001). (**D**) RPE-1 cells depleted with ADSL or control siRNAs were treated with vehicle or MRT00252040 and stained as in (**B, C**). Graph depicts the percentage of cells presenting two CP110 foci per centrosome (n = 2; scored 177 cells for siC + DMSO, 180 cells for siADSL + DMSO, 75 cells for siC + MRT00252040, 78 cells for siADSL + MRT00252040, *p<0.05).

*Figure 5 continued on next page*

*Figure 5 continued*

(**E**) RPE-1 cells depleted with ADSL and/or CP110 (silenced for 24 hr with three different siRNAs) were serum starved for 48 hr, fixed, and stained for ARL13B (magenta) and pericentrin (PCNT) (green). Graphs show the number of ciliated cells (n = 3 for siC, siADSL, siCP110#1, siADSL + siCP110#1; n = 2 for siCP110#2, siCP110#3, siADSL + siCP110#2 and siADSL + siCP110#3, scored 461 cells for siC, 301 cells for siADSL, 277 cells for siCP110#1, 289 cells for siADSL + siCP110#1, 119 cells for siCP110#2, 90 cells for siADSL + siCP110#2, 141 cells for siCP110#3, 98 cells for siADSL + siCP110#3, ****p<0.0001, ***p<0.001) and cilia length (n = 3, one-way ANOVA ***p<0.001, **p<0.01, *p<0.05). All graphs show means ± SD with individual values shown in circles.

The online version of this article includes the following source data and figure supplement(s) for figure 5:

**Source data 1.** Related to *Figure 5*.

**Figure supplement 1.** Analysis of pericentrin (PCNT) accumulation and CP110 depletion.

**Figure supplement 1—source data 1.** Related to *Figure 5—figure supplement 1A and C*.

with brain development. Alcian Blue staining showed that nearly 50% of the Adsl-depleted embryos exhibited weak or absent staining (*Figure 6F*). Defects in skull formation could be largely rescued by zebrafish *Adsl* or human *ADSL* expression but not expression of a human ADSL R426H mutant, the most frequently observed ADSLD mutation (*Figure 6F*; https://www.adenylosuccinatelyasedeficiency.com/). Examination of DNA damage signaling in the developing neural tube revealed an increase in γH2AX-positive cells. Similar to what was observed in RPE-1 cells, treatment with nucleosides suppressed DNA damage signaling (*Figure 6G*). These data demonstrated that Adsl depletion strongly impaired normal zebrafish development, leading to DNA damage that could be suppressed with nucleoside supplementation and several phenotypes consistent with ciliary defects.

## Adsl depletion impairs ciliogenesis in zebrafish

As the observed phenotypes were potentially indicative of defects in cilium function, we examined heart looping by staining for cardiac myosin light chain 2 (*cmcl2*) mRNA. Adsl-depleted embryos showed higher frequencies of defects, including inverse looping and to a lesser extent no loops (*Figure 7A*). Inverse heart looping may be indicative of laterality impairment (situs inversus) that can arise due to ciliary defects. To corroborate this possibility, we examined liver placement by staining for angiopoietin-like 3 (*angptl3*). A significant increase in inverse liver placement was observed in Adsl-depleted embryos compared to controls, supporting a general defect in laterality (*Figure 7B*). To further investigate the laterality defects, we examined left-right asymmetry at the 20-somite stage, staining for the mRNA of the left lateral plate mesoderm marker *southpaw* (*spaw*). Consistent with the altered distribution of *cmcl2* and *angptl3*, asymmetric *spaw* mRNA localization was changed in about 40% of Adsl-depleted embryos. Most of these embryos showed symmetric patterning and a smaller fraction of no or only weak staining. The correct asymmetric distribution of *spaw* mRNA could be largely restored by expression of mRNA encoding zebrafish Adsl (*Figure 7C*).

To test if impaired laterality may involve ciliary defects, we examined the Kupffer's vesicle (KV, organ of laterality). While KV area and cilia number were not significantly affected by Adsl depletion, cilia length was reduced in Adsl ATG MO-treated embryos, a phenotype that was partially rescued by co-injection of RNA encoding zebrafish Adsl (*Figure 7D–G*). These data, in combination with additional phenotypes, including laterality defects and hydrocephalus, support the role of ADSL in promoting proper cilia formation or function in vivo.

## MTX treatment rescues neurogenesis in Adsl-depleted zebrafish

As ciliogenesis defects were linked to impaired DNPS and excess SAICAr in human cells (*Figure 4*), we examined the effects of inhibiting the DNPS pathway during zebrafish development. We quantified the effects of Adsl depletion on differentiating neuronal cells by staining for the marker Elavl3/4. Similar to what we observed in the chicken neural tube, depletion of *Adsl* caused a significant reduction in Elavl3/4-positive cells that could be rescued by the co-injection of RNA encoding zebrafish Adsl (*Figure 8A*). We next treated control and Adsl-depleted embryos with MTX to attenuate the DNPS pathway upstream of ADSL and reduce SAICAR production. Treatment with MTX completely rescued the reduction in Elavl3/4-positive cells in the neural tube, indicating that this was not a result of impaired DNPS per se, but likely a consequence of intermediate metabolite accumulation (*Figure 8B*). Similar results were observed with a second morpholino targeting Adsl (*Figure 8—figure supplement 1*). We next examined the effect of MTX treatment on Sox2-positive neural progenitors in the developing

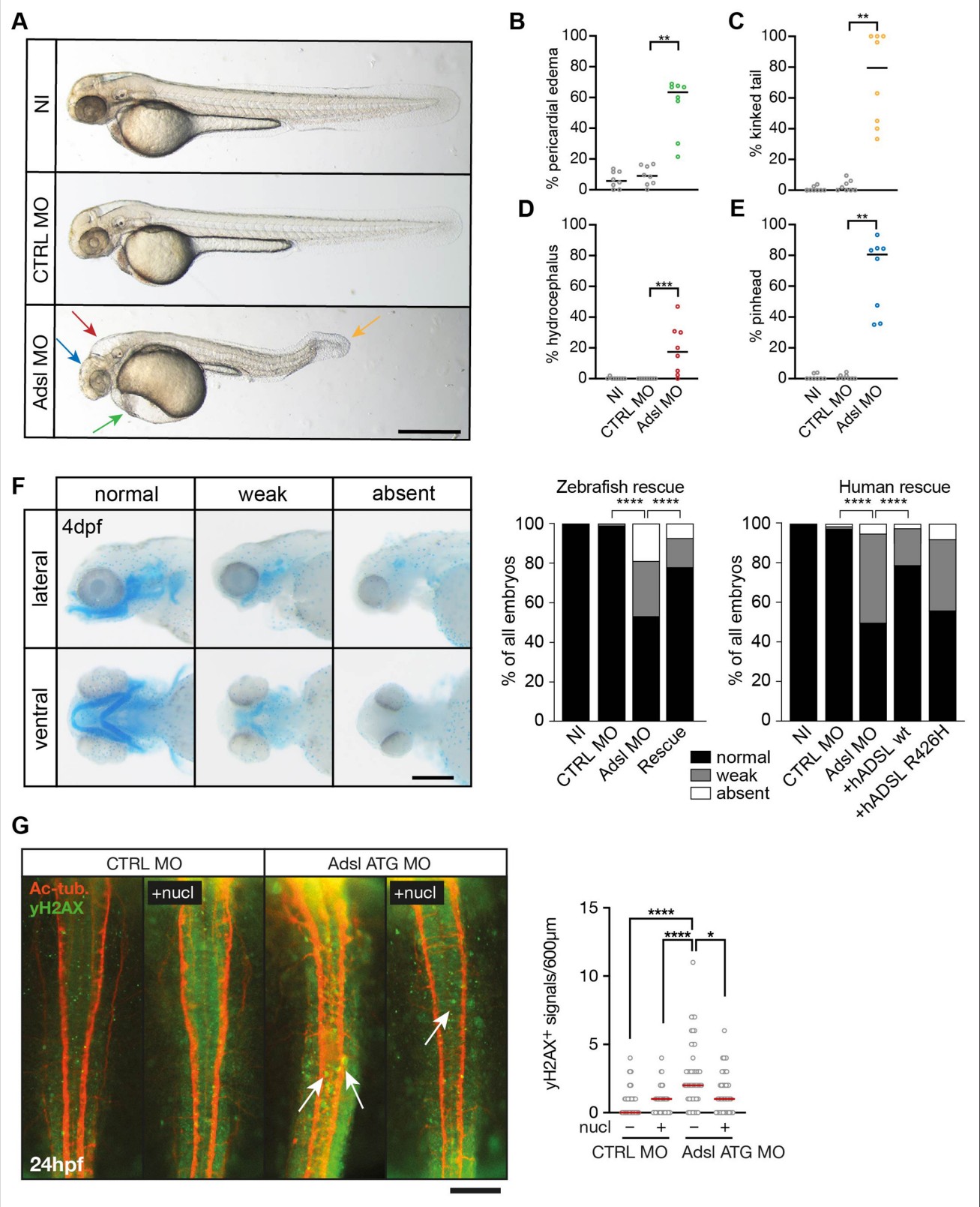

**Figure 6.** Depletion of Adsl in zebrafish causes developmental phenotypes and DNA damage signaling. (**A**) Live images of 48 hr post fertilization (hpf) zebrafish embryos showing pericardial edema (green arrow), kinked tail (yellow arrow), hydrocephalus (red arrow), and pinhead (blue arrow). NI (uninjected controls), CTRL MO (embryos injected with a standard control MO), Adsl ATG MO (injected with a translation blocking MO against Adsl). Scale bar = 500 μm. (**B–E**) Quantification of the percentage of embryos developing the indicated phenotypes. For (**B–E**), each circle indicates

*Figure 6 continued on next page*

*Figure 6 continued*

one experiment. Data from eight experiments with 311 embryos (NI), 275 (CTRL MO), and 227 (Adsl ATG MO) is shown. Kruskal–Wallis test with Dunn's multiple comparison. Dashes show median. **p=0.0042 (pericardial edema), **p=0.0032 (kinked tail), **p=0.0011 (pinhead), ***p=0.0005 (hydrocephalus). (**F**) Adsl-depleted zebrafish display skull formation defects. Cartilage staining of zebrafish embryos (4 days post fertilization [dpf]) with Alcian Blue. Embryos were classified according to the severity of their phenotype in normal staining, weak staining, or absent cartilage. Lateral and ventral view. Cartilage formation could be rescued by co-injection of capped mRNA encoding zebrafish Adsl. 6–8 experiments with a total of 178 embryos (NI), 133 (CTRL MO), 169 (Adsl ATG MO), and 123 (Rescue). Injection of mRNA encoding human wt ADSL, but not the R426H ADSLD variant, restores cartilage formation in embryos. Four experiments with a total of 116 embryos (NI), 81 (CTRL MO), 80 (Adsl ATG MO), 91 (+ *hADSL* wt), and 89 (+ *hADSL* R426H). Two-tailed Fisher's exact test; ****p<0.0001. Scale bar = 200 μm. (**G**) Immunofluorescence staining of the neural tube (dorsal view) of control and Adsl-depleted embryos 24 hpf for γH2AX (green) and acetylated-tubulin (Ac-tub: red). Treatment with 60 mM nucleosides was carried out in indicated samples. Experiments with 45 embryos per treatment are shown, dashes indicate median. Data were analyzed by using Kruskal–Wallis test with Dunn's correction. *p<0.05, ****p<0.0001. Scale bar = 300 μm. Unless indicated, comparisons are not significant.

The online version of this article includes the following source data and figure supplement(s) for figure 6:

**Source data 1.** Related to *Figure 6A–E*.

**Source data 2.** Related to *Figure 6F*.

**Source data 3.** Related to *Figure 6F*.

**Source data 4.** Related to *Figure 6G*.

**Figure supplement 1.** *Adsl* expression in zebrafish development.

**Figure supplement 2.** Test of knockdown efficiency.

forebrain (*Figure 8C*, upper right panel). Depletion of Adsl reduced the number of Sox2-positive cells, and this was rescued by co-treatment with MTX (*Figure 8C*). In contrast, supplementation with nucleosides did not rescue the reduced number of Sox2-positive progenitors in Adsl-depleted embryos (*Figure 8D*). These data indicate a specific role of impaired DNPS, and likely SAICAr accumulation, in the neural progenitor defects associated with Adsl depletion.

## Discussion

Despite a detailed understanding of the enzymology of DNPS and the purine nucleotide cycle, the specific cell and organismal effects underlying the complex etiology of ADSLD remain unclear. Our results uncovered multiple phenotypes associated with ADSL depletion in human RPE-1 cells, as well as developing chicken and zebrafish embryos (Appendix 1). These can largely be rescued by interventions that suppress the generation of metabolites in the DNPS pathway or restore purine levels, complementing defective DNPS or purine nucleotide pathway activity. DNA damage signaling was suppressed by nucleoside supplementation in human cells and zebrafish, suggesting that this was caused by purine deficiency (*Figure 1A*). In contrast, defects in primary ciliogenesis resulting from ADSL depletion were rescued by PAICS inhibitor or MTX treatment, but not nucleosides, and phenocopied by SAICAr administration. The inability to rescue these phenotypes with nucleoside supplementation indicated that they most likely resulted specifically from loss of ADSL, and potentially SAICAr accumulation, rather than a general deficiency in purine supply. As SAICAr levels were barely detectable in RPE-1, it could indicate that a very small amount of SAICAr is sufficient to provoke the phenotypes or that signaling elicited specifically from loss of the ADSL protein is responsible. In ADSL-depleted RPE-1 cells, we could also detect p53 activation and defects in cell cycle progression, in the absence of cell death or senescence, and these phenotypes were insensitive to nucleoside supplementation or DNPS modulation, indicating the involvement of additional pathways.

Microcephaly, which is present in a subset of more severe ADSLD cases (*Jurecka et al., 2008*; *Jurecka et al., 2012*; *Mouchegh et al., 2007*), was observed in zebrafish embryos following Adsl depletion. Similarly, ADSL depletion in chicken embryos led to a reduction in neural tube size. Together, the results suggest that they are potentially valuable models for understanding the etiology of microcephaly in ADSLD (*Jurecka et al., 2015*). DNA damage, p53 activation, and defects in cilia function, which we observed following ADSL depletion, have all been implicated in neurodevelopmental disorders, such as Seckel syndrome, which is associated with microcephaly (*Stracker et al., 2020*). In Seckel syndrome mice expressing hypomorphic, humanized alleles of *Atr*, replication stress and DNA damage preceded neuroprogenitor cell death (*O'Driscoll et al., 2003*; *Murga et al., 2009*). This was accompanied by extensive p53 activation, but co-deletion of p53 exacerbated the cellular

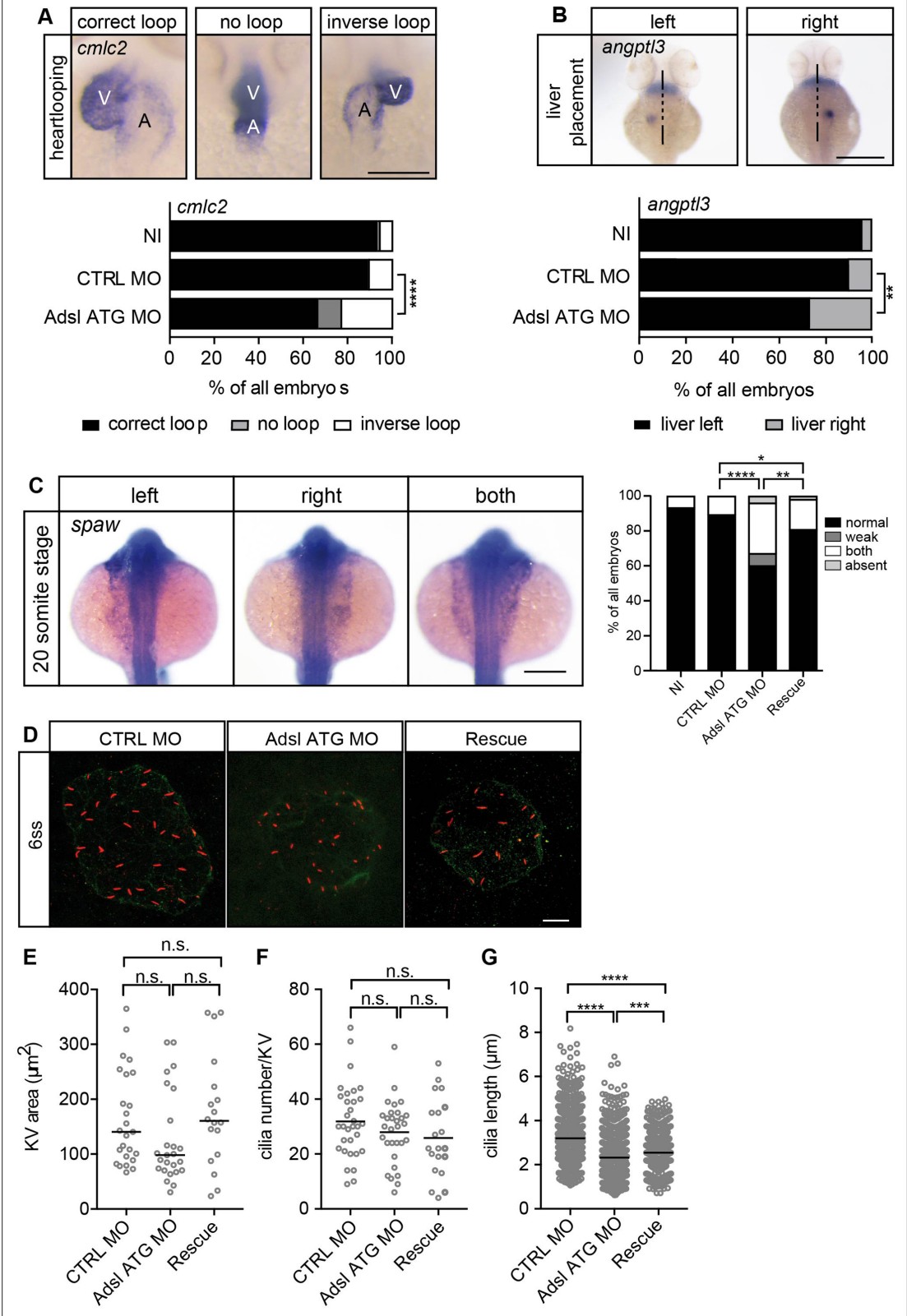

**Figure 7.** Impaired left-right (LR) asymmetry and cilium formation in the organ of laterality. (**A**) At 48 hr post fertilization (hpf), the ventricle (V) of the two-chambered zebrafish heart is placed left and above the atrium (**A**). Adsl-depleted embryos more frequently develop inversely looped hearts or developed unlooped hearts (no loop) (as scored by whole-mount in situ hybridization for *cardiac myosin light chain 2* [*cmlc2*]). N = 6 experiments with a total of 266 embryos (NI), 176 embryos (CTRL MO), and 188 embryos (Adsl ATG MO). Scale bar = 100 µm. (**B**) Whole-mount in situ hybridization for

*Figure 7 continued on next page*

*Figure 7 continued*

*angiopoietin-like 3* (*angptl3*) to assess liver placement in 48 hpf embryos. Dorsal view. Scale bar = 200 μm. 185 NI, 121 CTRL MO, and 99 Adsl ATG MO embryos. (**A, B**) Two-tailed Fisher's exact test; \**p<0.0015, \*\*\*\*p<0.0001. (**C**) Whole-mount in situ hybridization for the left lateral plate mesoderm marker *southpaw* (*spaw*) at 20 somite stage (ss). *Spaw* is normally expressed in the left lateral plate mesoderm. When LR asymmetry is disturbed, *spaw* can be detected on the right side or on both sides. Aberrant expression of *spaw* in Adsl morphants. Co-injection of RNA encoding zebrafish Adsl restores proper *spaw* expression. Two-tailed Fisher's exact test; \*p=0.0451, \*\*p=0.0016, \*\*\*\*p<0.0001. Results from five experiments with 121 embryos (NI), 142 (CTRL MO), 128 (Adsl ATG MO), and 105 (Rescue) are shown. Scale bar = 200 μm. (**D**) Confocal z-stacks of the Kupffer's vesicle (KV) of six ss embryos. Cilia are stained red (acetylated tubulin), while apical cell borders were stained for PKC ζ (green). Scale bar = 10 μm. (**E**) No significant changes in the size of the KV upon Adsl depletion. n = 25 (CTRL MO), 25 (Adsl ATG MO), and 18 embryos (rescue with zebrafish *adsl* RNA). Each circle is one embryo, line indicates median. Kruskal–Wallis test with Dunn's correction. p-values: CTRL MO vs. Adsl ATG MO: 0.2582; CTRL MO vs. Rescue: >0.9999; Adsl ATG MO vs. Rescue: 0.1684. (**F**) No significant changes in the number of cilia per KV. n = 32 (CTRL MO), 30 (Adsl ATG MO), and 20 embryos (rescue with zebrafish *adsl* RNA). Each circle is one embryo, lines show means. One-way ANOVA with Sidak's multiple comparison test. p=0.5538 (CTRL MO vs. Adsl ATG MO), 0.2844 (CTRL MO vs. Rescue), and 0.9225 (Adsl ATG MO vs. Rescue). (**G**) Shorter cilia in Adsl morphants can be partially elongated by co-injection of RNA encoding zebrafish Adsl. n = 960 cilia (CTRL MO), 798 (Adsl ATG MO), and 540 (Rescue). Kruskal–Wallis test with Dunn's correction, lines indicate medians; \*\*\*p=0.0008.

The online version of this article includes the following source data for figure 7:

**Source data 1.** Related to *Figure 7A and B*.

**Source data 2.** Related to *Figure 7C*.

**Source data 3.** Related to *Figure 7E*.

**Source data 4.** Related to *Figure 7F*.

**Source data 5.** Related to *Figure 7G*.

and organismal phenotypes, indicating a protective effect of p53 induction. Mutations in centrosomal proteins, such as CENPJ/SAS4/CPAP or CEP63, were also implicated in Seckel syndrome (*Al-Dosari et al., 2010*; *Sir et al., 2011*). In contrast to mice expressing hypomorphic *Atr*, progenitor loss in mice with CEP63 or CENPJ/SAS4 deficiency resulted from mitotic delays and the activation of the USP28-53BP1-dependent mitotic surveillance pathway (*Bazzi and Anderson, 2014*; *Insolera et al., 2014*; *Marjanović et al., 2015*; *McIntyre et al., 2012*; *Lin et al., 2020*; *Fong et al., 2016*; *Lambrus et al., 2016*). In this case, the phenotype was completely rescued by p53 co-deletion, revealing p53-dependent cell death as a main driver of the phenotype. Despite some phenotypic similarities with Seckel syndrome at the cellular level, including replication stress and p53 activation, we did not detect increased cell death as a result of ADSL depletion. This indicated that the reduced cellularity caused by ADSL depletion occurs in a manner mechanistically distinct from Seckel syndrome driven by ATR deficiency or centrosome duplication defects.

GMPS (GMP synthase), an essential enzyme in the purine synthesis pathway (*Figure 1—figure supplement 1*), was identified as a target for p53 repression following p53 activation using Nutlin-3a (*Holzer et al., 2017*). In addition, GMPS was demonstrated to promote p53 stabilization in response to genotoxic stress in a manner that requires the USP7 ubiquitin protease (*Reddy et al., 2014*). While we cannot rule out a role for this pathway in p53 stabilization following ADSL depletion, we could not rescue p53 activation with nucleoside supplementation and induction of p53 via the GMPS pathway was demonstrated to induce cell death, which we did not observe clearly in any of the systems we analyzed.

In addition to p53-dependent and -independent cell death, defects in cilium removal and premature progenitor differentiation have also been clearly implicated in microcephaly in Seckel syndrome and primary microcephaly (*Gabriel et al., 2016*; *Farooq et al., 2020*). We examined the removal of cilia by adding back serum to serum-starved RPE-1 cells depleted for ADSL and observed normal rates of cilia removal (*Figure 4—figure supplement 1G*), suggesting that impaired cilia resorption was likely not impairing cell cycle exit and indirectly impacting ciliogenesis. Analysis of differentiation in RPE-1 cells depleted for ADSL also did not support a premature differentiation phenotype (*Figure 1—figure supplement 2K*), but we recognize that RPE-1 cells may not respond to purine depletion in the same way as neural progenitors or other cell types in vivo.

In RPE-1 and chicken embryos, we observed cell cycle delays following ADSL depletion, as well as an overall reduction in SOX2-positive progenitor cells in both in vivo systems. As a result, differentiated ELAVL3/4-positive cell numbers were also reduced in both chicken and fish systems. Cell cycle delay was further supported by the observation that, contrary to overall Sox2-positive cell

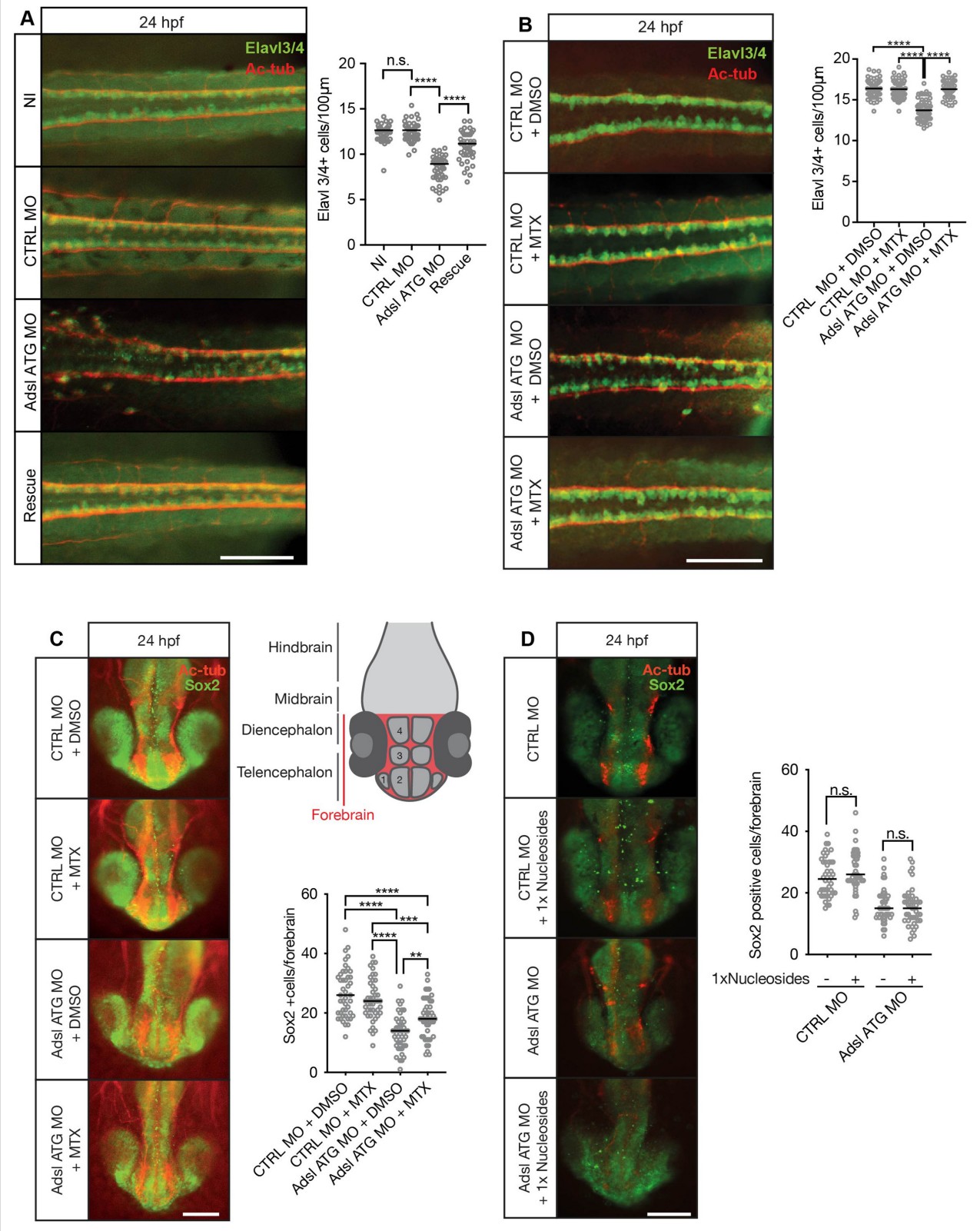

**Figure 8.** Adsl depletion reduces neuronal lineage cell numbers that can be rescued by methotrexate (MTX) treatment. (**A**) Immunofluorescence whole-mount microscopy of neural tubes of 24 hr post fertilization (hpf) zebrafish embryos (dorsal view) stained for acetylated tubulin (axons, red) and Elavl3/4 (green). Fewer Elavl3/4-positive cells in Adsl-depleted embryos that can be rescued by co-injection with RNA encoding zebrafish Adsl (Rescue). Graph shows Elavl3/4 counts of individual embryos, line indicates median. Three experiments with 45 embryos (NI), 45 (CTRL MO), 45 (Adsl ATG MO), and 45

*Figure 8 continued on next page*

*Figure 8 continued*

(Rescue). Kruskal–Wallis test with Dunn's correction. *ns*, not significant. p>0.9999, ****p<0.0001. Scale bar = 100 µm. (**B**) MTX treatment rescues Elavl3/4-positive cell numbers. Staining of the neural tube (dorsal view) of 24 hpf zebrafish embryos for acetylated tubulin (red) and or Elavl3/4 (green). Adsl morphants show fewer Elavl3/4-positive cells, which could be rescued by treatment with 100 µM MTX. Five experiments with 69 (CTRL MO), 75 (CTRL MO + MTX), 63 (Adsl ATG MO), and 58 (Adsl ATG MO + MTX) embryos. One-way ANOVA with Sidak's multiple comparison. ns p>0.9999, ****p<0.0001. Scale bar = 100 µm. (**C**) Forebrains of 24 hpf zebrafish embryos (left panels) stained for acetylated tubulin (red) and Sox2-positive neural progenitors (green), anterior view. Scale bar = 200 µm. Schematic of the developing brain of zebrafish embryos adapted from *Vaz et al., 2019*, top-right panel. The forebrain (red) is composed of the telencephalon with the olfactory bulb (1), the pallium (2), the optic recess region (3), and the diencephalon with the hypothalamus (4). Quantification of phenotypes (bottom-right panel). Adsl morphants show fewer neural progenitor cells in the forebrain, a defect that can partially be rescued with 100 µM MTX from tailbud stage on. Data were analyzed using one-way ANOVA with Sidak's multiple comparison. Dashes show medians. Experiments with 45 embryos (CTRL MO + DMSO), 45 embryos (CTRL MO + MTX), 45 embryos (Adsl ATG MO + DMSO), and 47 embryos (Adsl ATG MO + MTX). If not shown in the graph, all other comparisons are not significant. (**D**) Forebrains of 24 hr post fertilization (hpf) zebrafish embryos (left panels) stained for acetylated tubulin (red) and Sox2-positive neural progenitors (green), anterior view. Scale bar = 100 µm. Quantification of phenotypes (right panel). Adsl morphants show similar neural progenitor cells in the forebrain regardless of supplementation with 1× nucleosides. Data were analyzed using one-way ANOVA with Sidak's multiple comparison, n.s. = p>0.9999. Dashes show medians. Experiments with 45 embryos (CTRL MO + DMSO), 45 embryos (CTRL MO + MTX), 45 embryos (Adsl ATG MO + DMSO), and 47 embryos (Adsl ATG MO + MTX).

The online version of this article includes the following source data and figure supplement(s) for figure 8:

**Source data 1.** Related to *Figure 8A*.

**Source data 2.** Related to *Figure 8B*.

**Source data 3.** Related to *Figure 8C*.

**Source data 4.** Related to *Figure 8D*.

**Figure supplement 1.** Dorsal view of part of the neural tube of 24 hr post fertilization (hpf) zebrafish embryos (left panels).

**Figure supplement 1—source data 1.** Related to *Figure 8—figure supplement 1*.

numbers, the number of transfected, Sox2-positive cells in chicken embryos was increased relative to controls. In agreement with our findings, nutrient restriction was shown to arrest the proliferation of neural progenitors of *Xenopus larvae* and zebrafish reversibly in G2, suggesting that most of the cells were quiescent (*McKeown and Cline, 2019*), consistent with the lack of evidence for cell death or senescence in ADSL-depleted systems. While arrest in *Xenopus* progenitors did not require mTOR signaling, nutrient-dependent cell cycle reentry was mTOR dependent. Purine deficiency inhibits the mTORC1 pathway, which regulates protein synthesis in response to nutrient availability (*Hoxhaj et al., 2017*), and inactivation of mTORC1 results in microcephaly in mouse models, but this was attributed mainly to increased levels of cell death (*Cloëtta et al., 2013*). Thus, while we cannot rule out a role for dysregulated mTOR signaling in the phenotypes associated with ADSLD, the origin of microcephaly is likely to be mechanistically distinct to that in mTORC1-deficient mouse models. We therefore propose that the reduction in brain size resulting from ADSL depletion is largely due to the impaired cell cycle progression observed in SOX2-positive progenitors and delayed differentiation.

The molecular targets of SAICAR remain largely unclear, but its accumulation has been linked to activation of PKM2, and other kinases, in the context of glucose deficiency in cancer (*Keller et al., 2012*; *Keller et al., 2014*; *Yan et al., 2016*). Recent work also implicated ADSL in the activation of MYC, which plays a major role in controlling metabolism and proliferation in cancer cells (*Zurlo et al., 2019*). Considering that ciliogenesis is a highly regulated process and tightly coordinated with the cell cycle, modulation of central signaling pathways, including AKT and mTOR that control energetics and proliferation, may be one way by which SAICAr could affect cilia (*Hoxhaj et al., 2017*; *Walia et al., 2019*). In particular, AKT was shown to regulate a key step in ciliary vesicle formation and AKT activation exerts an inhibitory effect on ciliogenesis induced by serum starvation (*Walia et al., 2019*). We examined AKT activation following SAICAr treatment and did not observe any detectable increase, indicating that the ability of SAICAr to inhibit ciliogenesis occurs through a distinct mechanism, consistent with the fact that CP110 depletion, which occurs prior to vesicle formation, rescued ciliation in ADSL-depleted cells (*Figure 5E*). ADSL has also been identified as a proximal interactor of numerous centrosomal proteins, including CEP104, CEP128, CEP135, CEP152, CEP89, and centrobin, suggesting that it may play specific regulatory roles involving direct interactions or its enzymatic activity locally at centrosomes (*Gupta et al., 2015*). To address this, future work will be needed to determine the cellular interactomes of ADSL and SAICAr, and the signaling they elicit, to identify the disease-relevant targets.

To our knowledge, this is the first demonstration of a specific purine metabolite impairing ciliogenesis. While ciliopathy-like features have not been described for the pathology of ADSLD, we observed a robust rescue of the neural progenitor population in zebrafish following MTX treatment, suggesting that SAICAr and its effects on cilia may be involved. Consistent with effects on cilia in vivo, we observed shorter cilia in the KV of zebrafish, as well as several ciliopathy-related phenotypes (*Figures 6A–F , and 7A–G*), consistent with impairment of ciliogenesis or cilia function in vivo. Adsl-depleted zebrafish also presented with defects in skull cartilage formation that is coordinated with brain size, in part through cilia-based sonic hedgehog signaling (*Xavier et al., 2016*). Primary cilia and Hedgehog signaling have well-established roles in regulating multiple progenitor populations in the developing brain (*Bangs and Anderson, 2017*). However, we also note that severe defects in primary cilia function or Hedgehog signaling cause more drastic reductions in progenitor proliferation and numbers than we observed in either system, potentially consistent with the milder ciliogenesis effects observed in vivo (*Breunig et al., 2008*; *Han et al., 2008*). Moreover, we used ADSL knockdown in our experimental systems, and it is currently unclear to what extent the observed ciliary defects would be recapitulated by ADSL mutations in patients and in what tissues and cell types. If ADSL deficiency would be less severe or only a subset of tissues would be affected, patients may not present classic and widespread ciliopathy features. Together, our work provides a cell-level analysis of ADSL deficiency, identifies specific cellular defects, and ascribes these to defects in specific steps of DNPS or the purine nucleotide cycle. Highlighting the complex etiology of ADSLD, our results add further support to the notion that SAICAr may play a key role in pathological outcomes and establish a framework for deciphering the underlying molecular mechanisms.

# Materials and methods

## Key resources table

| Reagent type (species) or resource | Designation | Source or reference | Identifiers | Additional information |
|---|---|---|---|---|
| Gene (*Homo sapiens*) | *ADSL isoform1* | NCBI Gene | CCDS14001.1 | |
| Gene (*Gallus gallus*) | *Adsl* | GenBank | M37901.1 | |
| Gene (*Danio rerio*) | *adsl* | GenBank | NM_199899.2 | |
| Gene (*D. rerio*) | *angptl3* | GenBank | AF379604 | |
| Gene (*D. rerio*) | *spaw* | GenBank | NM_180967 | |
| Gene (*D. rerio*) | *cmlc2* | GenBank | PMID:10491254 | |
| Strain, strain background (*Gallus gallus*, eggs) | White leghorn fertilized eggs | Granja Gilbert S.A. | | |
| Strain, strain background (*D. rerio*, male and female) | AB | ZFIN | ZFIN ID: ZDB-GENO-960809-7 | |
| Strain, strain background (*D. rerio*, male and female) | EK | ZFIN | ZFIN ID: ZDB-GENO-990520-2 | |
| Cell line (human) | hTERT-RPE-1 | ATCC | Cat# CRL-4000; RRID:CVCL_4388 | Authenticated by STR testing, routinely tested for mycoplasma and found negative |
| Cell line (human) | HeLa | ATCC | Cat# CCL-2; RRID:CVCL_0030 | Authenticated by STR testing, routinely tested for mycoplasma and found negative |
| Cell line (human) | hTERT-RPE-1 p53KO | Kind gift from Bryan Tsou, Sloan-Kettering Institute | PMID:27371829 | P53 genotype confirmed by western blotting, routinely tested for mycoplasma and found negative. |
| Cell line (human) | AD-293 | Agilent | Cat# 240085; RRID:CVCL_9804 | Authenticated by STR testing, routinely tested for mycoplasma and found negative. |
| Transfected construct (human) | pLenti-CMV-GFP-BLAST (659-1) | Addgene | Cat# 17445; RRID:Addgene_17445 | pLenti CMV GFP Blast (659-1) was a gift from Eric Campeau and Paul Kaufman (Addgene plasmid # 17445; http://n2t.net/addgene:17445; RRID:Addgene_17445) |
| Transfected construct (human) | pMDLg/pRRE | Addgene | Cat# 12251; RRID:Addgene_12251 | pMDLg/pRRE was a gift from Didier Trono (Addgene plasmid # 12251; http://n2t.net/addgene:12251; RRID:Addgene_12251) |

*Continued on next page*

*Continued*

| Reagent type (species) or resource | Designation | Source or reference | Identifiers | Additional information |
|---|---|---|---|---|
| Transfected construct (human) | pCMV-VSV-G | Addgene | Cat# 8454; RRID:Addgene_8454 | pCMV-VSV-G was a gift from Bob Weinberg (Addgene plasmid # 8454; http://n2t.net/addgene:8454; RRID:Addgene_8454) |
| Transfected construct (human) | pRSV-REV | Addgene | Cat# 12253; RRID:Addgene_12253 | pRSV-Rev was a gift from Didier Trono (Addgene plasmid # 12253; http://n2t.net/addgene:12253; RRID:Addgene_12253) |
| Antibody | Anti-acetylated α-tubulin (6-11B-1) (mouse monoclonal) | Santa Cruz Biotechnology, Inc | Cat# sc-23950; RRID:AB_628409 | IF (1:1000 human cells) |
| Antibody | Anti-Elav3/4 (rabbit polyclonal) | GeneTex | Cat# GTX128365; RRID:AB_2885764 | IF (1:1000 human cells) |
| Antibody | Anti-SOX2 (rabbit polyclonal) | Abcam | Cat# ab97959; RRID:AB_2341193 | IF (1:1000 human cells) |
| Antibody | Anti-γH2AX (rabbit polyclonal) | GeneTex | Cat# GTX127342; RRID:AB_2833105 | IF (1:400 human cells) |
| Antibody | Anti-AKT (pan) (rabbit polyclonal) | Cell Signaling | Cat# 2920; RRID:AB_1147620 | Western (1:2000 human cells) |
| Antibody | Anti-AKT p-S473 (rabbit polyclonal) | Cell Signaling | Cat# 4060; RRID:AB_2315049 | Western (1:2000 human cells) |
| Antibody | Anti-vinculin (rabbit polyclonal) | Cell Signaling | Cat# 13901; RRID:AB_2728768 | Western (1:2000 human cells) |
| Antibody | Anti-ADSL (rabbit polyclonal) | MilliporeSigma | Cat# HPA000525; RRID:AB_1078106 | IF (1:200 fish, 1:100 human cells) |
| Antibody | Anti-PKC ζ (C-20) (rabbit polyclonal) | Santa Cruz Biotechnology, Inc | Cat# sc-216; RRID:AB_2300359 | IF (1:500 fish) |
| Antibody | Alexa Fluor 568 donkey anti-mouse IgG (donkey polyclonal) | Molecular Probes | Cat# A10037; RRID:AB_2534013 | IF (1:1000 fish) |
| Antibody | Alexa Fluor 488 donkey anti-rabbit IgG (donkey polyclonal) | Molecular Probes | Cat# A32790; RRID:AB_2762833 | IF (1:1000 fish) |
| Antibody | Alexa Fluor 594 goat anti-mouse IgG cross-adsorbed (goat polyclonal) | Invitrogen/ Thermo Fisher Scientific | Cat# A32742; RRID:AB_2762825 | IF (1:400 human cells) |
| Antibody | Alexa Fluor 488 goat anti-rabbit IgG cross-adsorbed (goat polyclonal) | Invitrogen/ Thermo Fisher Scientific | Cat# A11034; RRID:AB_2576217 | IF (1:400 human cells) |
| Antibody | Anti-digoxigenin-AP Fab fragments (sheep polyclonal) | Roche | Cat# 11093274910; RRID:AB_514497 | ISH (1:5000 fish) |
| Antibody | Anti-actin (AC-40) | MilliporeSigma | Cat# A4700; RRID:AB_476730 | Western (1:1500 human cells) |
| Antibody | Anti-Ki67 (mouse monoclonal) | Novocastra | Cat# NCL-Ki-67p; RRID:AB_442102 | IF (1:500 human cells) |
| Antibody | Anti-p53 (1C12) | Cell Signaling | Cat# 2524; RRID:AB_331743 | IF (1:100 human cells) |
| Antibody | Anti-vimentin (rabbit polyclonal) | Abcam | Cat# ab45939; RRID:AB_2257290 | IF (1:100 human cells) |
| Antibody | Anti-CK20 (Ks20.8) (mouse monoclonal) | Dako/Agilent | Cat# GA77761-2 | IF (1:200 human cells) |
| Antibody | Anti-53BP1 (rabbit polyclonal) | Novus Biologicals | Cat# NB100-304; RRID:AB_10003037 | IF (1:400 human cells) |
| Antibody | Anti-γH2AX-pS139 (rabbit polyclonal) | Santa Cruz Biotechnology, Inc | Cat# sc-517336; RRID:AB_2133718 | IF (1:100 human cells) |
| Antibody | Anti-RPA32 | MilliporeSigma | Cat# NA19L; RRID:AB_565123 | IF (1:100 human cells) |
| Antibody | Anti-SOX2 (rabbit polyclonal) | Invitrogen/ Thermo Fisher Scientific | Cat# 48-1400; RRID:AB_2533841 | IF (1:500 chicken) |

*Continued*

| Reagent type (species) or resource | Designation | Source or reference | Identifiers | Additional information |
|---|---|---|---|---|
| Antibody | Anti-ELAVL3/4 (HuC/HuD) 16A11 | Molecular Probes/ Thermo Fisher Scientific | Cat# A21271; RRID:AB_221448 | IF (1:500 chicken) |
| Antibody | Anti-pH3S10 (rabbit polyclonal) | MilliporeSigma | Cat# 06-570; RRID:AB_310177 | IF (1:500 chicken) |
| Antibody | Anti-cleaved caspase-3 (CC3) | MilliporeSigma | Cat# AB3623; RRID:AB_91556 | IF (1:500 chicken) |
| Antibody | Anti-TUJ1 | Covance | Cat# MMS-435P; RRID:AB_2313773 | IF (1:1000 chicken) |
| Antibody | PAX6 (mouse monoclonal) | DSHB | Cat# AB_528427; RRID:AB_528427 | IF (1:250 chicken) |
| Antibody | Anti-ARL13B (C5) (mouse monoclonal) | Santa Cruz Biotechnology | Cat# sc-515784; RRID:AB_2890034 | IF (1:100) |
| Antibody | Anti-pericentrin (PCNT) (rabbit polyclonal) | Novus Biologicals | Cat# NBP1-87772; RRID:AB_11018354 | IF (1:400) |
| Antibody | Anti-CP110 (rabbit polyclonal) | Kind gift from Andrew Holland | Unpublished reagent | IF (1:1000) |
| Antibody | Anti-centrin (20H5) (mouse monoclonal) | MilliporeSigma | Cat# 04-1624; RRID:AB_AB_10563501 | IF (1:1000) |
| Antibody | Anti-centrobin (mouse monoclonal) | Kind gift from Ciaran Morrison | PMID:29440264 | IF (1:500) |
| Recombinant DNA reagent | pLenti-CMV-ADSL*-EGFP siRNA resistant | This paper | | |
| Recombinant DNA reagent | pCR2.1-TOPO | Invitrogen/ Thermo Fisher Scientific | Cat# 450640; RRID:Addgene_26778 | |
| Recombinant DNA reagent | pCRII-zfAdsl | This paper | | Template for generation of antisense in situ probe (zebrafish) |
| Recombinant DNA reagent | pCS2+ Flag-zfAdsl | This paper | | Template for generation of capped mRNA (zebrafish) |
| Recombinant DNA reagent | pCS2+ Flag-zfAdsl MO mut | This paper | | Template for generation of capped mRNA (insensitive to ATG MO) (zebrafish) |
| Recombinant DNA reagent | pCS2+ hAdsl | This paper | | Template for generation of capped mRNA (zebrafish) |
| Recombinant DNA reagent | pCS2+ hAdslR426H | This paper | | Template for generation of capped mRNA (zebrafish) |
| Recombinant DNA reagent | pSHIN | Kind gift of Dr. Kojima | PMID:24741441 | |
| Sequence-based reagent | ADSL Smartpool siRNA | Dharmacon | Cat# M-010986-01-0005 | |
| Sequence-based reagent | ADSL siRNA#2 | MilliporeSigma | Custom | CAAGAUUUGCACCGACAUA |
| Sequence-based reagent | CP110 siRNA#1 | MilliporeSigma | Custom | GCAAAACCAGAAUACGAGAUU |
| Sequence-based reagent | CP110 siRNA#2 | MilliporeSigma | Custom | CAAGCGGACUCACUCCAUATT |
| Sequence-based reagent | CP110 siRNA#3 | MilliporeSigma | Custom | TAGACTTATGCAGACAGATAA |
| Sequence-based reagent | EGFP siRNA | MilliporeSigma | Custom | GGCUACGUCCAGGAGCGCCGCACC |
| Sequence-based reagent | GL2 siRNA (siC, targets luciferase) | MilliporeSigma | Custom, published in PMID:11373684 | CGUACGCGGAAUACUUCGA |
| Sequence-based reagent | ADSL-BsiWI-F | MilliporeSigma | Custom | 5'AAAACGTACGATGG CGGCTGGAGGCGATCAT3' |

*Continued on next page*

*Continued*

| Reagent type (species) or resource | Designation | Source or reference | Identifiers | Additional information |
|---|---|---|---|---|
| Sequence-based reagent | ADSL-*EcoR1-R* | MilliporeSigma | Custom | 5'TTTTGAATTCCAGACA TAATTCTGCTTTCA3' |
| Sequence-based reagent | shCONTROL | MilliporeSigma | Custom, for control in chicken embryo | 5'-CCGGTCTCGA CGGTCGAGT-3' |
| Sequence-based reagent | shADSL | MilliporeSigma | Custom, for ADSL depletion in chicken embryo | 5'-GAGCTGGACA GATTAGTGA-3' |
| Sequence-based reagent | Adsl ATG MO | GeneTools | Custom, for Adsl depletion in fish | 5'-TCCCTCCATGC CTGCAGC GGTTAAA |
| Sequence-based reagent | Adsl splMO | GeneTools | Custom, for Adsl depletion in fish | 5'-CCAACTGTGGG AGAGAGC GACTGTA |
| Sequence-based reagent | Std. CTRL MO | GeneTools | Custom, for control in fish | 5'-CCTCTTACCTCA GTTACAATTTATA |
| Commercial assay or kit | Click-iT EdU imaging kit | Invitrogen/ Thermo Fisher Scientific | Cat# C10340 | |
| Commercial assay or kit | TOPO TA cloning | Invitrogen/ Thermo Fisher Scientific | Cat# 450640 | |
| Commercial assay or kit | AmpliCapTM SP6 High Yield Message Maker Kit | Cellscript | Cat# C-AC0706 | |
| Commercial assay or kit | QuikChange Lightning mutagenesis kit | Thermo Fisher Scientific | Cat# 210518 | |
| Commercial assay or kit | DC protein assay | Bio-Rad | Cat# 500-0111 | |
| Commercial assay or kit | Immobilion ECL Ultra | MilliporeSigma | Cat# WBULS0100 | |
| Chemical compound, drug | Formaldehyde, 37% | MilliporeSigma | Cat# 47608-500ML-F | Used at 4% for cell fixation |
| Commercial assay or kit | Senescence β-Galactosidase Staining Kit | Cell Signaling | Cat# 9860 | |
| Commercial assay or kit | High Capacity RNA-to-cDNA Kit | Applied Biosystems | Cat# 4387406 | |
| Commercial assay or kit | TaqMan Universal PCR Master Mix | Thermo Fisher Scientific | Cat# 4324018 | |
| Commercial assay or kit | Quick ligation kit | NEB | Cat# 4324018 | |
| Chemical compound, drug | Lipofectamine RNAiMAX | Thermo Fisher Scientific | Cat# 13778150 | |
| Chemical compound, drug | Opti-MEM | Gibco | Cat# 31985070 | |
| Chemical compound, drug | DMEM-F12 | Gibco | Cat# 21331046 | |
| Chemical compound, drug | Tri-reagent | MilliporeSigma | Cat# T2494 | |
| Chemical compound, drug | KOD Hot start DNA polymerase | MilliporeSigma | Cat# 71086-3 | |
| Chemical compound, drug | Polyethylenimine (PEI), linear (MW 25,000) | Polyscience Euro | Cat# 23966-2 | |
| Chemical compound, drug | Bovine serum albumin | MilliporeSigma | Cat# F7524 | 10% in culture media |
| Chemical compound, drug | Trypan Blue | Gibco/ Thermo Fisher Scientific | Cat# 15250061 | 0.4% solution |
| Chemical compound, drug | Triton X-100 (TX-100) | MilliporeSigma | Cat# T8787 | |

*Continued on next page*

*Continued*

| Reagent type (species) or resource | Designation | Source or reference | Identifiers | Additional information |
|---|---|---|---|---|
| Chemical compound, drug | Phosphatase inhibitor cocktail 2 | MilliporeSigma | Cat# P5726-5ML | 1× in lysis buffer |
| Chemical compound, drug | Phosphatase inhibitor cocktail 3 | MilliporeSigma | Cat# P0044-5ML | 1× in lysis buffer |
| Chemical compound, drug | cOmplete, EDTA free Protease inhibitors | Roche/MilliporeSigma | Cat# 4693132001 | 1× in lysis buffer |
| Chemical compound, drug | Alcian Blue Solution | Sigma | Cat# B8438 | |
| Chemical compound, drug | MTX (methotrexate) | Cayman Chemical | Cat# 13960 | 100 µM in DMSO |
| Chemical compound, drug | EmbryoMax Nucleosides 100× | Merck | Cat# ES-008-D | 1× in media or saline |
| Chemical compound, drug | TWEEN 20 | Sigma | Cat# P2287 | |
| Chemical compound, drug | MRT00252040 | Kindly provided by Simon Osborne, LifeArc, London, UK | | PAICS inhibitor Stock solution 2 mM in DMSO, used at final concentration of 2 µM in DMSO |
| Chemical compound, drug | Methotrexate (MTX) | MilliporeSigma | Cat# M8407-100MG | DHFR inhibitor Stock solution of 100 µM and used at final concentration of 4 µM in DMSO |
| Chemical compound, drug | KU-55933 | Selleckchem | Cat# 118500-2MG | ATM inhibitor Used at final concentration of 5 µM in DMSO in cell culture media |
| Chemical compound, drug | Penicillin–streptomycin (10K U/ml) | Thermo Fisher Scientific | Cat# 15140122 | Used 1% in cell culture media |
| Chemical compound, drug | SAICAr | CarboSynth | Cat# NS16860 | Stock solution 20 mg/ml in water and used as final concentration at 1 mg/ml in cell culture media |
| Chemical compound, drug | Doxorubicin | MilliporeSigma | Cat# D1515 | Used at 1 µg/ml in cell culture media |
| Chemical compound, drug | AscI | NEB | Cat# R0558S | |
| Chemical compound, drug | NotI-HF | NEB | Cat# R3189S | |
| Chemical compound, drug | Carbenicillin | MilliporeSigma | Cat# C9231-1G | |
| Chemical compound, drug | Blasticidin | Invitrogen/ Thermo Fisher Scientific | Cat# A1113902 | |
| Chemical compound, drug | Paraformaldehyde (PFA) | PanReac AppliChem | Cat# 14145.1211 | Used at 2% in cells |
| Software, algorithm | GraphPad Prism 7–9 | GraphPad Prism | RRID:SCR_002798 | Version: 7.0e, 8.4.3, 9.3.0 |
| Software, algorithm | Adobe Photoshop | Adobe Photoshop | RRID:SCR_014199 | Version: 22.4.2 |
| Software, algorithm | Adobe Illustrator | Adobe Illustrator | RRID:SCR_010279 | Version: 24.3, 25.2.3 |
| Software, algorithm | Fiji | Fiji | RRID:SCR_002285 | Version: 2.0.0-rc-69/1.52p |
| Software, algorithm | LAS AF | Leica | RRID:SCR_013673 | Version: 3.7.3.23245 |

## Human cell culture

Human immortalized hTERT-RPE-1 WT (ATCC), TP53 knockout (kind gift from Brian Tsou), RPE-1-expressing pLenti-EGFP and pLenti-ADSL*EGFP (siRNA-resistant mutant) cells were cultured in Dulbecco's modified Eagle medium-F12 (DMEM-F12; Thermo Fisher Scientific) supplemented with 10% (v/v) fetal bovine serum (MilliporeSigma) and 100 U/ml penicillin–streptomycin at 37°C and 5% $CO_2$ in humidified atmosphere. For cilia experiments, silenced RPE-1 cells were serum starved for 48 hr in OptiMEM (Thermo Fisher Scientific). Hela (ATCC) and AD293 (Agilent) cells were cultured in DMEM with high glucose (Thermo Fisher Scientific) supplemented with 10% (v/v) fetal bovine serum (MilliporeSigma) and 100 U/ml penicillin–streptomycin at 37°C and 5% $CO_2$ in humidified atmosphere.

All cell lines were authenticated by STR testing (ATCC) and routinely tested for mycoplasma and found negative (Universal Mycoplasma Detection Kit; ATCC).

## Drugs used and concentrations

1 mg/ml SAICAr (Carbo Synth) was added to the cells for 96 hr to mimic ADSL depletion. 60 µM nucleosides (100X Embryomax, Merck Millipore) were added from the first silencing to the end at 1× in the culture medium. MRT00252040 (kindly provided by Simon Osborne, LifeArc, London, UK) dissolved in DMSO was used at 2 µM and MTX (MilliporeSigma) at 4 µM as described in *Hoxhaj et al., 2017*. ATM inhibitor (KU-55933; Selleckchem) was used at 5 mM for 24 hr before fixation. Doxorubicin (MilliporeSigma) was used as positive control for senescence at 1 µg/ml for 6 days. Adenosine (MilliporeSigma) was dissolved in water at a concentration of 0.8 mg/ml and diluted 100× to reach a final concentration of 80 µg/ml for 96 hr. HU (from MilliporeSigma) has been freshly dissolved in water and used at a final concentration of 2 mM in cell culture for 6 hr.

## Antibodies

Staining of human cells was performed with the following primary antibodies: α-ADSL (MilliporeSigma, rabbit, 1:100 IF, 1:1000 western), α-ARL13B (Santa Cruz Biotechnology, mouse monoclonal C5, 1:100), PCNT (Novus Biologicals, rabbit, 1:400), α-p53 (Cell Signaling, mouse monoclonal 1C12, 1:100), α–RPA2 (Calbiochem, mouse monoclonal Ab-3, 1:100), a-53BP1 (Novus Biologicals, rabbit, 1:400), α-pSer139-H2A.X (Santa Cruz Biotechnology, rabbit, 1:100), α-AKT (Cell Signaling, rabbit polyclonal, 1:2000), α-pAKT-S473 (Cell Signaling, rabbit polyclonal, 1:2000), α-vinculin (Cell Signaling, rabbit polyclonal, 1:2000), α-actin (MilliporeSigma, mouse monoclonal AC-40, 1:1500), α-vimentin (Abcam, rabbit, 1:100), α-CK20 (DaKo, mouse, 1:200), α-Ki67 (Novocastra, mouse, 1:500), α-centrobin (a kind gift from Ciaran Morrison, mouse, 1:500 [*Ogungbenro et al., 2018*]), α-centrin (EMD Millipore, mouse, 1:1000), α-CP110 (a kind gift from Andrew Holland, rabbit, 1:1000), α-mouse-Alexa Fluor 594 (Molecular Probes, goat, 1:400), and α-rabbit-Alexa Fluor 488 (Molecular Probes, goat, 1:400). Staining of chicken tissues was performed with the following primary antibodies: α-ELAVL3/4 (Molecular Probes, mouse, 1:500), α-β-TubulinIII-Tuj1 (Covance, mouse, 1:1000), Pax6 (DSHB, mouse, 1:250), SOX2 (Invitrogen, rabbit, 1:500), pH3S10 (Millipore, rabbit, 1:500), and cleaved caspase-3 (Millipore, rabbit, 1:500). Staining of zebrafish tissues was performed with the following primary and secondary antibodies: α-ELAVL3/4 (GeneTex, rabbit, 1:1000), α-acetylated-alpha-tubulin (Santa Cruz Biotechnology, mouse monoclonal 6-11B-1, 1:1000), α-SOX2 (Abcam, rabbit, 1:1000), α-γH2AX (GeneTex, rabbit, 1:400), α-ADSL (MilliporeSigma, rabbit, 1:200), α-PKC ζ (Santa Cruz Biotechnology, rabbit, 1:500), α-digoxigenin-AP Fab fragments (Roche, sheep, 1:5000 ISH), α-mouse-Alexa Fluor 568 (Molecular Probes, donkey, 1:1000), and α-rabbit-Alexa Fluor 488 (Molecular Probes, donkey, 1:1000).

## siRNA transfections

RPE-1 (hTERT-RPE-1; ATCC) were transfected with 100 nM siRNAs (MilliporeSigma or Dharmacon) with Lipofectamine RNAiMAX (Thermo Fisher Scientific) in Opti-MEM (Gibco) without antibiotics for one or two rounds of 48 hr, depending on the gene to be silenced. We used siGFP (GGCUACGU CCAGGAGCGCCGCACC) and siGL2 (siC) (CGUACGCGGAAUACUUCGA) as negative controls. In this study, we used a smart pool (four siRNAs) against *ADSL* (Dharmacon) or single oligos siADSL#2 5′-CAAGAUUUGCACCGACAUA-3′ (MilliporeSigma). The siRNA-resistant mutant was designed to be resistant to siADSL#2. For rescue experiments with siCP110, we used three oligos (#1 5′-GCAAA ACCAGAAUACGAGAUU-3′, #2 5′-CAAGCGGACUCACUCCAUATT-3′, and #3 5′- TAGACTTATGCA GACAGATAA-3′ [MilliporeSigma] for 24 hr).

## RNA extraction and quantitative real-time PCR

RPE-1 cells (ATCC) were seeded in a six-well plate, silenced for 96 hr, washed twice in PBS, and resuspended in 300 µl of Tri-Reagent (MilliporeSigma). RNA was isolated by centrifugation followed by chloroform extraction, isopropanol precipitation, washing twice in 75% ethanol, and resuspended in 20 µl DEPC-treated water (Thermo Fisher Scientific). Total RNA was quantified with a Nanodrop 8000 Instrument (Thermo Fisher Scientific). 1 µg of total RNA was used for the reverse transcription reaction performed by High-Capacity RNA-to-cDNA Kit (Applied Biosystems), according to the manufacturer's recommendations, in a 2× RT buffer mix, supplemented with dNTPs, random primers, and RT enzyme

in a final volume of 20 µl. Quantitative real-time PCR (qRT-PCR) was performed using the comparative CT method and a Step-One-Plus Real-Time PCR Instrument (Thermo Fisher). Amplification of the 16 ng of cDNA was done in triplicate with TaqMan Universal PCR Master Mix (Thermo Fisher) for *ADSL* and *GAPDH*.

## Plasmid cloning and generation of stable cell line

The siRNA-resistant mutant, ADSL*, was produced by introducing five different silent mutations using the QuikChange mutagenesis kit (Thermo Fisher) with the following primers: forward, 5'-GGTTT GCCAGGAGGCGTAGGTCTTTGCAAATTGTGTGCACTGATGCCCCCA-3'; reverse, 5'-CCAAACGGT CCTCCGCATCCAGAAACGTTTAACACACGTGACT ACGGGGGT-3'. Constructs were checked by sequencing (Macrogen), and expression was checked by Western blot and immunofluorescence into the pLenti-CMV-eGFP-BLAST (659-1) plasmid, a gift from Eric Campeau and Paul Kaufman (Addgene plasmid #17445; http://n2t.net/addgene:17445; RRID:Addgene_17445) (*Campeau et al., 2009*) using the primers containing *Xho*I and *Eco*RI overhangs (ADSL-*Xho*I forward 5'-AAAACTCGAGCGA TGGCGGCTGGAGGCGATCAT-3' and ADSL-*Eco*RI reverse 5'-TTTTGAATTCCAGACATAATTCTGC TTCA-3'). The siRNA-resistant mutant was produced by introducing five different silent mutations using the QuikChange mutagenesis kit (Thermo Fisher) with the following primers: forward, 5'-GGTTT GCCAGGAGGCGTAGGTCTTTGCAAATTGTGTGCACTGATGCCCCCA-3'; reverse, 5'-CCAAACGGT CCTCCGCATCCAGAAACGTTTAACACACGTGACT ACGGGGGT-3'. Constructs were checked by sequencing (Macrogen), and expression was checked by Western blot and immunofluorescence. For lentivirus preparation: 6 × 10⁶ AD293 cells were plated in 15 cm culture dishes and transfected with 20 µg pLenti-CMV-EGFP empty and pLenti-CMV-ADSL*-EGFP, 2 µg pRSV-REV, a gift from Didier Trono (Addgene plasmid # 12253; http://n2t.net/addgene:12253; RRID:Addgene_12253), 6 µg pMDLg/pRRE, a gift from Didier Trono (Addgene plasmid # 12251; http://n2t.net/addgene:12251; RRID:Addgene_12251), and 2 µg pCMV-VSV-G, a gift from Bob Weinberg (Addgene plasmid # 8454; http://n2t.net/addgene:8454; RRID:Addgene_8454), plasmids with 160 µl PEI pH 7.0 (Polyscience Euro), and 150 mM NaCl (*Dull et al., 1998*; *Stewart et al., 2003*). After 48 hr, the medium containing the viruses was cleared with a 0.45 mm filter (Millipore) and added to the target cells. Three days after the infection, cells were selected with blasticidin (Invitrogen) for 7 days.

## Immunofluorescence (human cells)

Silenced RPE-1 cells were seeded on 18 mm round coverslips after 96 hr of silencing and fixed accordingly with the antibody requirements, with 4% paraformaldehyde (PFA) for 10 or 30 min, followed by permeabilization in 0.1% Triton-PBS for 5 min and stored in 100% EtOH. For RPA2 and γH2AX detection, pre-extraction was performed using cold 0.2% Triton X-100 in 1× PBS on ice for 5 min, before fixation and permeabilization as described before. Cells were incubated with the blocking solution of 3% bovine serum albumin (MilliporeSigma) in PBT for 30 min. Primary antibodies (listed below) were diluted in the same blocking solution and incubated for 1 hr at room temperature (RT). After three washes, cells were incubated with Alexa Fluor-conjugated 594 and 488 secondary antibodies (Thermo Fisher Scientific) at 1:400 dilution for 1 hr at RT. DAPI was used to visualize the DNA. Slides were imaged using Orca AG camera (Hamamatsu) on a Leica DMI6000B microscope equipped with 1.4 100× oil immersion objective. AF6000 software (Leica) was used for image acquisition. Image processing and quantification was performed with ImageJ software. Intensities were measured in images acquired with the same exposure settings and subtracting the background for each image.

## Cell proliferation and cell death

150,000 RPE-1 cells were plated in six-well plates and silenced with control or siADSL oligos (MilliporeSigma or Dharmacon) for 72 hr, when they were counted and plated again in the same amount for the second round of silencing. After 3 days, cells were counted as second timepoint (144 hr, 6 days) and seeded for a third timepoint (9 days). Cells were cultured in the presence of serum for all the experiments. The ΔPDL (difference in population doubling levels) was obtained by using the formula: log(N1/N0)/log2, where N1 is the number of cells at the timepoint we collected them and N0 is the initial number of cells plated (*Pantoja and Serrano, 1999*). For detecting cell death, cells in suspension were collected in the growth medium and the attached ones were trypsinized and resuspended

in complete medium to block trypsin activity. Cells were then mixed in 0.4% Trypan Blue solution (Gibco). The number of blue-positive cells and total cell number was quantified at the microscope.

## Cell cycle profile

$10 * 10^6$ cells were collected after 96 hr of siRNA depletion, as previously described, and fixed in cold 100% EtOH dropwise and stored at –20°C for 24 hr. Cells were stained with propidium iodide (MilliporeSigma) in PBS and RNase, and 10,000 cells were analyzed at the Parc Cientific de Barcelona flow cytometry facility using a Gallios (Beckman Coulter) instrument. The data were collected and analyzed using FlowJo v10.7.2.

## Cell extracts and Western blotting

RPE-1 cells were seeded in a six-well plate, and after 96 hr of silencing they were trypsinized, washed once in PBS, and resuspended in a 2× SDS lysis buffer (2× SDS lysis buffer contained 4% SDS, 20% glycerol, 120 mM Tris/HCl pH 6.8, 1X protease [Roche], and phosphatase inhibitors [MilliporeSigma]). Protein concentration was quantified using the *DC* Protein Assay (Bio-Rad), and proteins separated by SDS-PAGE and transferred to 0.2 µm nitrocellulose membrane (Amersham Protran) or 0.45 µm PVDF membrane (MilliporeSigma) depending on the molecular weight. Membranes were blocked in 5% milk in PBT (PBS containing 0.2% Tween-20) for 30 min and then incubated with primary antibodies for 1 hr at RT. After three washes in PBS containing Tween-20 0.02%, membranes were incubated with secondary antibodies conjugated to HRP and protein bands were visualized by ECL-Plus (MilliporeSigma).

## Senescence-associated (SA) β-galactosidase assay

RPE-1 were silenced for 96 hr with siControl and siADSL#2, then fixed in ice-cold X-gal fixative solution (containing 4% formaldehyde, 0.5% glutaraldehyde, 0.1 M sodium phosphate buffer pH 7.2) for 4 min. After two washes in PBS, X-gal (Roche) was diluted 1:100 at a final concentration of 1 mg/ml in X-gal solution (containing 5 mM $K_3Fe(CN)_6$, 5 mM $K_4Fe(CN)_6$, 2 mM $MgCl_2$ in PBS). Incubation was performed at 37°C for 8 hr in the dark. Two washes in PBS were performed before taking the images. Doxorubicin was used as a positive control.

## Targeted liquid chromatography-mass spectrometry (LC-MS) analyses

For *Figure 1C* and *Figure 1—figure supplement 2A*, metabolites were extracted from snap-frozen cell pellets by adding 300 µl ACN:MeOH:$H_2O$ (5:4:1, v:v:v) solution and vortexing samples for 30 s. Samples were immersed in liquid $N_2$ to disrupt cell membranes followed by 30 s of ultrasonication. These two steps were repeated three times. Then, samples were incubated at –20°C for 1 hr, centrifuged at 15,200 rpm for 10 min at 4°C, and the supernatant was collected into an LC-MS vial.

For cell media, metabolites were extracted from 200 µl of lyophilized media by adding 300 µl acetyonitrile:MeOH:$H_2O$ (5:4:1, v:v:v) and vortexing samples for 30 s. Then, samples were incubated at –20°C for 20 min, centrifuged at 15,200 rpm for 10 min at 4°C, and the supernatant was collected into an LC-MS vial. Samples were injected (1 µl of media and 10 µl of cell extracts) in a UHPLC system (1290 Agilent) coupled to a triple quadrupole (QqQ) MS (6490 Agilent Technologies) operated in multiple reaction monitoring (MRM) and positive (POS) or negative (NEG) electrospray ionization (ESI) mode. Source parameters were gas temperature (°C) = 270; gas flow (l/min) = 15; nebulizer (psi) = 25; capillary (V) = 3500 (POS) and 2500 (NEG). Metabolites were separated using an ACQUITY UPLC BEH HILIC 1.7 µm, 2.1 × 150 mm chromatography (Waters) at a flow rate of 0.4 ml/min. The solvent system was A = 50 mM ammonium acetate in water and B = acetonitrile. The linear gradient elution started at 5% A (time 0–1 min), 50% A (time 1–4 min), 5% A (time 4–4.5 min), and finished at 5% A (time 7 min). MRM transitions are shown in Appendix 2.

The ribosides SAICAr and SAdo and the internal standards $^{13}C_4$-SAICAr and $^{13}C_4$-SAdo were synthesized according to previously published procedures (*Baresova et al., 2016*; *Zikánová et al., 2005*). SAICAr and S-Ado analysis was performed essentially as previously described (*Mádrová et al., 2018*). For *Figure 1—figure supplement 2B and C*, RPE-1 or HeLa cells were sonicated in 10 mM Tris pH 8,2, 2 mM EDTA, 10 mM KCl, 1 mM DTT, and 4% glycerol, centrifugated, metabolites extracted from 50 µl of lysate diluted to protein concentration 1 mg/ml with 200 µl of extracting solution (ACN:MeOH 1:1, v:v, 0.125 M formic acid, 15 nM $^{13}C_4$-SAICAr and $^{13}C_4$-SAdo), and incubated on dry ice for 30 min.

After centrifugation, the supernatant was deep-freeze-dried and pellets were resuspended in 50 µl of water, briefly centrifuged, and analyzed by LC-MS/MS.

The reverse-phase column Prontosil 120-3C18 AQ 150 × 3 mm (Bischoff Chromatography) and Agilent 1290 Infinity LC System coupled with an API 4000 triple quadrupole mass spectrometer with an electron ion source (Agilent Technologies) were used. The mobile phase consisted of 0.1% formic acid in water (mobile phase A) and 0.1% formic acid in acetonitrile (mobile phase B). The gradient elution was performed as follows: t = 0.0 min, 100% A; t = 2.5–6.5 min, linear gradient to 20% B; t = 6.5–7.5, linear gradient from 20% to 60% B; t = 7.5–9.0 min, 90% B and then a regeneration of the column for 12 min. The flow rate was set to 0.3 ml/min for the first step and increased to 0.4 ml/min. The injection volume was 5 µl. MRM transitions for SAICAr, SAdo, SAICAR, SAMP, and their corresponding internal standards are shown in Appendix 2.

The chromatographic conditions were the same as in the HPLC-HRMS$^n$ analysis mentioned above. Detection was performed on an Orbitrap Elite operating in positive ionization mode with the same setting as above. The detection method was divided into four time segments. Full scan analysis within the mass range $m/z$ 70–1000 was performed in the first (0.0–3.0 min) and fourth (12.0–17.0 min) segments. The selected ion monitoring (SIM) method was applied in the second segment (3.0–7.0 min) for the analysis of ribosides ($m/z$ 177–417) and in the third segment (7.0–12.0 min) for the analysis of ribotides ($m/z$ 257–497) to enhance the sensitivity towards these metabolites (except for the measurement of SAdo, which had $m/z$ ranges 379–389). The resolution was set to 60,000 FWHM. The mass error was below 3 ppm. All cell lines were measured in hexaplicate, and the intensity values are presented as averages. The identities of the accumulated compounds in both cell lysates and media were confirmed by MS$^2$ fragmentation analysis. Fragmentation spectra were produced via CID with the fragmentation energy set to 30 units of normalized collision energy.

## Statistical analysis (cells)

In vitro data were analyzed with an unpaired two-sided *t*-test when two samples were compared, while one-way ANOVA was used to compare more than two samples in the same graph (GraphPad Prism 6.0, GraphPad Software Inc). Values of $p < 0.05$ were considered statistically significant (*$p < 0.05$; **$p < 0.01$; ***$p < 0.001$; ****$p < 0.0001$). Two or more independent experiments were performed for each condition, and this is indicated in individual figure legends. Statistical analysis for percentages of positive cells: counts of cells belonging to the positive and negative condition were used to fit a generalized linear model with binomial distribution. The R [1] packages 'lme4' (*Jaeken and Van den Berghe, 1984*) and 'multcomp' (*Jurecka et al., 2015*) were used to fit the model and compute raw and adjusted p-values with the 'Shaffer' method.

## Cloning (fish)

To generate a template for the generation of an antisense in situ probe, a 921 bp fragment of the *Danio rerio adsl* open-reading frame was cloned into pCRII via TOPO TA cloning (Invitrogen). To have a template for the generation of capped mRNA using the AmpliCap SP6 High Yield Message Maker Kit (Cellscript), the whole open-reading frame of zebrafish *adsl* was cloned with an N-terminal Flag-tag into pCS2+ using *Eco*RI and *Xho*I.

## Immunofluorescence (fish)

Zebrafish embryos were fixed with 4% buffered PFA at the indicated stages. Antibody staining was performed as described (*Jaffe et al., 2010*) using the primary antibodies previously described (see also section 'Antibodies') and detected with Alexa Fluor-labeled secondary antibodies (1:1000, Molecular Probes).

## Statistical analysis (fish)

The number of fertilized eggs per clutch determined the size of experimental groups with clutches having been randomly and equally divided into treatment groups. No additional statistical methods have been applied to predetermine sample size. All zebrafish experiments were done at least three times with eggs from different mating tanks or different mating days. Embryo numbers are given in the legends. All statistical analyses were performed with GraphPad Prism 7 and 8, respectively. Data were tested for normality and analyzed accordingly by parametric or nonparametric tests. Graphs

display, if not indicated otherwise, individual datapoints and medians in case of nonparametric datasets. An α level of <0.5 was considered significant.

## Zebrafish maintenance and manipulation

Zebrafish were maintained in a 14 hr light and 10 hr dark cycle in a standardized, water recycling housing system (Tecniplast) with automatic monitoring and adjustments of pH, conductivity, and temperature. Fertilized eggs were generated by natural matings of the wild-type strains EK or AB. Eggs were incubated at 28.5°C and allowed to develop until the desired stages. In order to achieve Adsl knockdown, a translation blocking antisense MO (Adsl ATG MO) (5'-TCCCTCCATGCCTGCA GCGGTTAAA) was used or a MO that targets the exon-intron boundary at exon 4 of Adsl (Adsl SplMO) (5'-CCAACTGTGGGAGAGAGCGACTGTA). A standard control MO was also used in all experiments. MOs (GeneTools Inc) were injected at the 1–2-cell stage directly into the yolk. In addition, noninjected wild-type embryos served as internal control for clutch quality. For pharmacological manipulation, zebrafish embryos were immersed in embryo water containing 1% DMSO or 1% DMSO and 100 µM MTX (Cayman Chemical) from 10 until 24 hr post fertilization (hpf) or 50 µM nucleosides. All zebrafish maintenance and procedures have been approved by the Veterinary Care Unit at Ulm University and University of Tübingen, respectively, and the animal welfare commissioner of the regional board for scientific animal experiments in Tübingen, Germany. Zebrafish experiments were performed according to the European Union Directive 86/609/EEC for the protection of animals used for experimental and other scientific purposes.

## In situ hybridization (fish)

Zebrafish were fixed overnight at 4°C at the indicated stages using 4% buffered PFA, dehydrated with a gradual methanol series, and stored at –20°C until further use. For in situ hybridization (ISH), embryos were rehydrated in a methanol series containing PBST (PBS containing 0.1% Tween-20) and processed according to standard protocols (*Thisse and Thisse, 2014*). Genes of interest were detected using DIG-labeled in situ probes, which were in vitro transcribed from linearized plasmids carrying fragments of the gene of interest: *adsl* (GenBank no.199899.2), *angiopoietin-like 3* (*angptl3*, GenBank no. AF379604). The probes against *cardiac myosin light chain 2* (*cmcl2*) and *spaw* have been described before (*Burkhalter et al., 2013*).

## Analysis of cartilage formation

4 days post fertilization (dpf), old zebrafish embryos were fixed for 2 hr at RT using 4% buffered PFA. After rinsing with PBS, embryos were washed for 10 min with 50% EtOH in PBS before the staining solution (0.02% Alcian Blue [MilliporeSigma], 70% EtOH, 50 mM MgCl$_2$) was added, and the embryos were incubated overnight at RT. On the next day, embryos were rinsed with H$_2$O and subsequently bleached for 20 min at RT with opened lid of the reaction tube (bleaching solution: 1.5% H$_2$O$_2$ in 1% KOH). A clearing series was performed (30 min 20% glycerol/0.25% KOH, 2 hr 50% glycerol/0.1% KOH). Stained embryos were stored at 4°C in 50% glycerol/0.1% KOH.

## Measurements of cilia and neural progenitors/differentiated cell populations

To count neural progenitors, anterior views of 24 hpf embryos were taken using a fluorescent whole-mount microscope. The number of Sox2-positive cells within the forebrain was determined. To count differentiated neural cells, dorsal views of embryos were captured by fluorescent whole-mount microscopy and the number of ELAVL3/4-positive cells per 100 µm was counted. γH2AX-positive cells were counted over a distance of 300 µm in the neural tube. Cilia were counted and measured after acquiring confocal z-stacks of flat-mounted tails of six somite stage (ss) embryos. The Simple Neurite Tracer in Fiji was used to trace and measure cilia through the whole z-stack. ImageJ was also used to trace and measure the outline of the KVs.

## Microscopy of zebrafish embryos

Live zebrafish embryos and those processed by ISH or for cartilage staining were imaged using a M125 whole-mount microscope equipped with a Leica IC80 HD camera. Zebrafish embryos undergoing immunofluorescence stainings were assessed with an M205 FCA and a DFC 9000 GT sCMOS

camera. Confocal z-stacks were acquired on a TCS SP5II with LAS AF software (all microscopes and software: Leica).

## Chick embryo in ovo electroporation

Eggs from white leghorn chickens were incubated at 37.5°C in an atmosphere of 45% humidity, and the embryos were staged according to *Hamburger and Hamilton, 1951*. Chick embryos were electroporated with column-purified plasmid DNA (3 µg/µl for shRNAs) in $H_2O$ containing Fast Green (0.5 µg/µl). Briefly, plasmid DNA was injected into the lumen of HH12 or HH16 neural tubes, electrodes were placed on either side of the neural tube, and electroporation was carried out by applying five 50 ms square pulses using an Intracel Dual Pulse (TSS10) electroporator set at 25 V. Transfected embryos were allowed to develop to the specific stages and then dissected under a fluorescence dissection microscope.

## DNA constructs

shRNAs were generated using pSHIN plasmid (a GFP expressing evolution of pSUPER): shCONTROL sequence (CCGGTCTCGACGGTCGAGT) and shADSL sequence (GAGCTGGACAGATTAGTGA). The knockdown efficiency of shRNAs was assessed by RT-qPCR in electroporated chicken embryonic fibroblast cultures (*Herrera et al., 2014*).

## Immunostaining and EdU incorporation in chicken embryos

Embryos were fixed overnight at 4°C in 4% PFA, and immunostaining was performed on vibratome sections (60 µm) following standard procedures. After washing in PBS-0.1% Triton X-100, the sections were incubated overnight with the appropriate primary antibodies diluted in a solution of PBS-0.1% Triton supplemented with 10% bovine serum albumin. After washing in PBS-0.1% Triton, sections were incubated for 2 hr at RT with the appropriate Alexa-conjugated secondary antibodies diluted in a solution of PBS-0.1% Triton supplemented with 10% bovine serum albumin. After staining, the sections were mounted and examined on a Leica SP5 or a Zeiss Lsm 780 multiphoton microscope. For EdU incorporation, 200 µl of EdU solution (1 mM) was added on the vitelline membrane of each embryo 2 hr before fixation in 4% PFA. EdU was detected in sections using the Click-iT EdU imaging kit (Invitrogen).

## Fluorescence-associated cell sorting

HH-12 chicken embryos were electroporated with shCONTROL or shADSL plasmids, and 48 hr post electroporation (hpe), a single-cell suspension was obtained by digestion for 10–15 min with Trypsin-EDTA (MilliporeSigma) and labeled with Hoechst and α-ELAVL3/4 antibody used with Alexa 647-conjugaded anti-mouse secondary antibody. Alexa 647, Hoechst, and GFP fluorescence were determined by FACSAria Fusion cytometer (BD Biosciences), and the data were analyzed with FlowJo software (Tree Star) and Multicycle software (Phoenix Flow Systems; cell cycle profile analysis).

## Quantitative fluorescence image analysis

Quantification of cleaved caspase-3 immunofluorescence intensity was done using ImageJ software. Tuj1+ and Tuj1- areas on the electroporated side and the respective areas on the non-electroporated side were delimited by polygonal selection, and the mean intensity of cleaved caspase-3 immunofluorescence was quantified as mean gray values. At least three different images were used to calculate the mean value per embryo. Each mean value was normalized to the mean value obtained for the respective non-electroporated area of the same embryo.

## Acknowledgements

We thank the members of the Lüders, Roig and Stracker labs for input, A Riera for chemistry advice, C Morrison for Centrobin antibody, A Holland for CP110 antibody, B Tsou for p53-deficient RPE-1 cells, D Zafra for assistance, CSO Attolini and the IRB Barcelona Biostatistics and Bioinformatics Core Facility for statistical analysis, C Donow and S Burczyk for excellent help with zebrafish maintenance, and LifeArc for supplying MRT00252040. ID was funded by the European Union's Horizon 2020 research and innovation programme under the Marie Skłodowska-Curie grant agreement no. 754510, THS, JL, and SP were funded by the Ministry of Science, Innovation and Universities (MCIU;

PGC2018-095616-B-I00 to THS, PGC2018-099562-B-I00 to JL, and BFU2017-83562-P to SP), the 2017 SGR 1089 (AGAUR), FEDER, the Centres of Excellence Severo Ochoa award, and the CERCA Programme. THS was supported by the NIH Intramural Research Program, National Cancer Institute, Center for Cancer Research. MP was funded by grants from the Deutsche Forschungsgemeinschaft (DFG PH144/4-1 and PH144/6-1). MZ, OS, and VS were supported by Charles University, program PROGRES Q26/LF1. We would like to thank Biocev, First Faculty of Medicine, Charles University, for the opportunity to use their department's equipment.

## Additional information

### Funding

| Funder | Grant reference number | Author |
|---|---|---|
| H2020 Marie Skłodowska-Curie Actions | 754510 | Ilaria Dutto |
| Ministerio de Ciencia, Innovación y Universidades | PGC2018-099562-B-I00 | Jens Lüders |
| Ministerio de Ciencia, Innovación y Universidades | PGC2018-095616-B-I00 | Travis H Stracker |
| Deutsche Forschungsgemeinschaft | DFG PH144/4-1 | Melanie Philipp |
| Deutsche Forschungsgemeinschaft | PH144/6-1 | Melanie Philipp |
| Agència de Gestió d'Ajuts Universitaris i de Recerca | 2017 SGR | Jens Lüders<br>Travis H Stracker |
| Charles University | PROGRES Q26/LF1 | Olga Souckova<br>Václava Škopová<br>Marie Zikánová |
| Ministry of Science, Innovation and Universities | BFU2017-83562-P | Sebastian Pons |
| National Institutes of Health Clinical Center | Intramural Research Program | Travis H Stracker<br>Jordann A Smak |

The funders had no role in study design, data collection and interpretation, or the decision to submit the work for publication.

### Author contributions

Ilaria Dutto, Julian Gerhards, Conceptualization, Data curation, Formal analysis, Investigation, Methodology, Visualization, Writing – original draft, Writing – review and editing; Antonio Herrera, Data curation, Formal analysis, Investigation, Methodology, Visualization, Writing – original draft, Writing – review and editing; Olga Souckova, Václava Škopová, Jordann A Smak, Alexandra Junza, Martin D Burkhalter, Formal analysis, Investigation; Oscar Yanes, Data curation, Formal analysis, Investigation, Methodology; Cedric Boeckx, Conceptualization; Marie Zikánová, Formal analysis, Investigation, Methodology, Supervision, Writing – review and editing; Sebastian Pons, Formal analysis, Methodology, Resources, Supervision; Melanie Philipp, Data curation, Formal analysis, Funding acquisition, Investigation, Methodology, Supervision, Visualization, Writing – original draft, Writing – review and editing; Jens Lüders, Travis H Stracker, Conceptualization, Data curation, Formal analysis, Funding acquisition, Investigation, Methodology, Project administration, Resources, Supervision, Visualization, Writing – original draft, Writing – review and editing

### Author ORCIDs

Julian Gerhards ![ORCID] http://orcid.org/0000-0002-7005-1618
Antonio Herrera ![ORCID] http://orcid.org/0000-0002-6248-1001
Cedric Boeckx ![ORCID] http://orcid.org/0000-0001-8882-9718
Martin D Burkhalter ![ORCID] http://orcid.org/0000-0002-8646-3131

Melanie Philipp http://orcid.org/0000-0003-2714-965X
Jens Lüders http://orcid.org/0000-0002-9018-7977
Travis H Stracker http://orcid.org/0000-0002-8650-2081

### Decision letter and Author response
Decision letter https://doi.org/10.7554/eLife.70518.sa1
Author response https://doi.org/10.7554/eLife.70518.sa2

---

## Additional files

### Supplementary files
• Transparent reporting form

### Data availability
Most data generated or analysed during this study are included in the manuscript and supporting source data files. Additional source data is available via Figshare, https://doi.org/10.25452/figshare.plus.c.5793614.

The following dataset was generated:

| Author(s) | Year | Dataset title | Dataset URL | Database and Identifier |
|---|---|---|---|---|
| Stracker T, Lüders J, Dutto I, Philipp M, Pons S | 2022 | Source data supporting 'Pathway specific effects of ADSL deficiency on neurodevelopment' | https://doi.org/10.25452/figshare.plus.c.5793614 | Figshare, 10.25452/figshare.plus.c.5793614 |

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

# Appendix 1

Summary of phenotypes identified in the systems used. Corresponding figures are indicated, as well as interventions that rescue or recapitulate the phenotype.

**In vitro (RPE-1)**

| Phenotype | Defined by | Complemented with | Recapitulated by | Figures |
|---|---|---|---|---|
| Cell cycle exit | Ki67 positivity | ADSL expression, p53-KO | - | *Figure 1E, H, I and L, Figure 1—figure supplement 2I* |
| Cell cycle arrest (G1) | Increased P53 levels, flow cytometry, cumulative growth | P53 deletion | - | *Figure 1D, F, G, J and K, Figure 1—figure supplement 2E and H* |
| Elevated DNA damage signaling | γH2AX, 53BP1 foci | ATM inhibitor (53BP1), nucleosides, adenosine | Hydroxyurea, ionizing radiation | *Figure 2B–H* |
| Elevated replication stress | Chromatin bound RPA | Nucleosides, adenosine | - | *Figure 2A* |
| Ciliogenesis defect | IF analysis of ARL13B, centriole markers, CP110 | PAICS inhibition, MTX treatment, CP110 depletion | SAICAr addition | *Figure 4B–G, Figure 4—figure supplement 1C–E, Figure 5A–E* |

**In vivo (chicken)**

| Phenotype | Defined by | Complemented with | Recapitulated by | Figure |
|---|---|---|---|---|
| Microcephaly | Reduced neural tube size | - | - | *Figure 3B* |
| Cell cycle arrest (G2) | Edu/H3S10 IF, Flow cytometry | - | - | *Figure 3C–F* |
| Reduced Sox2+ progenitors | SOX2 IF, flow cytometry | - | - | *Figure 3B and C* |
| Reduced differentiated cells | ELAVL3/4 IF | - | - | *Figure 3B and C, Figure 3—figure supplement 1B* |

**In vivo (fish)**

| Phenotype | Defined by | Complemented with | Recapitulated by | Figure |
|---|---|---|---|---|
| Microcephaly | Pinhead, hydrocephalus, pericardial edema, kinked tail, defective skull formation | Fish or human ADSL expression | - | *Figure 6A–F* |
| Reduced Sox2+ progenitors | SOX2 IF | MTX treatment | - | *Figure 8B and C* |
| Reduced differentiated cells | ELAVL3/4 IF | MTX treatment, ADSL expression | - | *Figure 8A and B, Figure 8—figure supplement 1* |
| Ciliogenesis defect | Left-right patterning defect, reduced cilia length | ADSL expression | - | *Figure 6A–F, Figure 7A–G* |
| Elevated DNA damage signaling | gH2AX IF | Nucleosides | - | *Figure 6G* |

## Appendix 2

Multiple reaction monitoring transitions. See details in Materials and methods.

| Metabolite | First transition | Second transition | Polarity |
|---|---|---|---|
| GMP | 364→152 | 364→135 | Positive |
| AMP | 348→136 | 348→119 | Positive |
| SAICAr | 375→243 | 375→110 | Positive |
| SAdo | 384→252 | 384→234 | Positive |
| $^{13}C_4$-SAICAr | 379→247 | | Positive |
| $^{13}C_4$-SAdo | 388→256 | | Positive |

Multiple reaction monitoring transitions

