## [Editor Report]

This article attempts to elucidate adenylosuccinate lyase (ADSL) deficiency in neurodevelopment using a variety of approaches. It provides a better understanding of the underlying mechanisms that lead to microcephaly in some patients affected by ADSL deficiency.

---

## [Decision Letter]

[Editors' note: this paper was reviewed by Review Commons.]

---

## [Author Response]

Reviewer #1 (Evidence, reproducibility and clarity (Required)):The presented work by Stracker and Co attempts to elucidate ADSL deficiency in neurodevelopment using a variety of exciting approaches. The presented experiments are thorough, and an orthogonal experiment supports each finding. Overall, the manuscript is technically solid and brings new ideas and finding to the field. in vivo data with Chicken and Fish experiments of high quality. In my opinion, the manuscript requires are clarifications with minor sets of experiments as below.Authors claim that upon ADSL depletion, cells exit the cell cycle as well as p53 dependent arrest. Do they mean a proportion of cells have exited (G0), and the rest got stuck G2 to M transition via p53 dependent manner? I believe these two may not go well hand to hand. If so, it has to be dissected with proper quantifications.

Our analysis of cell cycle exit in RPE1 cells is based on Ki67 positivity (marker of cycling cells) in the population. To address this point, we performed flow cytometric analysis of cells using propidium iodide to allow the assessment of cell cycle profiles using DNA content. In wild type RPE1 cells, we observe a higher proportion of cells in G1 with a corresponding reduction of cells in S/G2 phase of the cell cycle, consistent with p53 induction driving a G1 arrest in fibroblasts. In contrast, this difference in cell cycle was not observed in p53 deficient RPE1 cells, indicating the arrest is p53 dependent following ADSL depletion. This data is included in new Figure 1—figure supplement E and H in the revised manuscript.

As we previously mentioned, we expected that RPE1 cells in culture may behave differently than NPCs in vivo. However, we believe the phenotype remains informative and demonstrates a clear impact of ADSL depletion on cell cycle progression, consistent with other data.

However, for me, it appears that these two phenomena are unrelated, as I could see from their Chicken experiment (which is done cleanly) that SOX2 positive cells get stuck at G2. Cell cycle exit would have caused depletion of NPCs, leading to NPCs’ overall reduction with the simultaneous increase in neurons. To prove their cell cycle exit theory, the authors should also conduct serum starvation and stimulation experiments.

As we previously indicated, we are not reporting that chicken NPCs are necessarily exiting the cell cycle in the ADSL depleted chicken embryos. NPCs arrest in G2, and we see slightly more Sox2+ NPCs and a consequent reduction in differentiated cells. We tried extensively to optimize p53 and Ki67 staining for use in the chicken samples to address this further but were unsuccessful.

We believe the NPCs in vivo are a better system for addressing the mechanisms underlying microcephaly but feel that reporting the RPE1 data is also important, as it may reflect defects in other cell types depleted for ADSL. We can also obtain more detail in RPE1s due to the nature of the in vitro system that is more amenable to controlled manipulation. We will clarify this point in the text related to several of the additional comments below.

We cannot perform the requested serum starvation/stimulation experiments in the NPCs in vivo, as it is not technically possible in this system, but we included this data for RPE1 cells in new Figure 4—figure supplement 1F.

Overall, It appears that the cells have G2 arrest predominantly, and cell cycle exit defect is probably not present (at least their data does not sufficiently prove it, see below).

We agree and again want to reiterate that while the 2 different systems both show a clear cell cycle progression phenotype, they do not behave in exactly the same way. They are different cell types, different organisms, and comparing the cultured cells to a subpopulation in vivo has many caveats that we indeed recognize (ex. constant serum stimulation in culture vs growth factor gradients and limitations in vivo). As mentioned, we tried extensively to optimize p53 and Ki67 for use in the chicken embryos, purchasing several different antibodies and working with our histochemistry core facility, but were unable to achieve staining that was reliable for quantification. We therefore are technically unable to address cell cycle exit in the chicken system but agree that the in vivo data suggests it is unlikely in NPCs (at least in the time frame of our analysis), a point we will address in the revised discussion.

Figure 4: Measuring cell cycle parameters will help assess whether there is an undisrupted cell cycle progression in the p53 KO lines.

As previously described, we carried out cell cycle analysis in RPE1 lines to examine the distribution following ADSL depletion and found that indeed, wild type cells arrested in G1 in a p53 dependent manner (new Figure 1—figure supplement 2E and H). However, we recognize that this may not directly inform about the potential role of p53 in vivo*.* This is unfortunately a limitation of the chicken embryo system.

Figure 5B should use cilia imaging to ensure that the authors look at the proper cell stage. Interestingly, SAICAr impairs the removal of CP110. The details are missing such that the concentrations used (at the figure panel itself).

In agreement with the literature, Figure 4A confirms that serum starvation causes essentially all cells to exit the cell cycle and produce a cilium. Thus, we can conclude that essentially all cells in Figure 5B are in G0. We have reviewed the legends and methods and added all pertinent details to both for clarity.

It is essential to perform this experiment in cycling cells to avoid the effect of serum starvation. Again, cell cycle parameters should be measured whether there is a defective proliferation (or G2 arrest).

We established in Figure 1—figure supplement 2E that there is not a G2 arrest in RPE-1, but rather a p53-dependent G1 arrest. As the reviewer is likely aware, serum starvation is used in this assay to drive cells into G0, which is permissive for cilium assembly. This allows us to compare ciliogenesis in control and ADSL knockdown cells, independently of any cell cycle effects (in cycling cells cilia are only transiently assembled in G1 and only in roughly 20% of cells in RPE1). We feel that performing this experiment in cycling cells will not be very informative due to the low percentage of ciliated cells and the fact that cell cycle defects and DNA damage signaling in ADSL-depleted cells will further complicate the analysis.

However, we examined cilia disassembly following serum add back in RPE-1 (roughly as performed in (Gabriel et al., 2016)) and we did not observe a defect in timely disassembly, as has been implicated in NPC attrition in Seckel Syndrome. We have included this data in Figure 4—figure supplement 1F and indicated this point in the text and discussion.

The experiments in Figure 5 are very intriguing and bring valuable insights. Numerous graphs without representative figures make it hard to interpret. I recommend putting the findings in a tabular form and provide respective imaging data.

We added additional representative data for experimental conditions to the new Figure 5. As many of the graphs are repeats with different siRNAs to control for off target effects, we did not add representative panels for each condition to avoid crowding the figure, as the effects of the CP110 siRNAs are fairly uniform regardless of the siRNA used.

In the Discussion section, maybe the authors can help me get it straight. ADSL depletion accompanied by cilia loss is suggested to elevate the cellular level of SAICAr. In their orthogonal experiment, adding the very same compound impairs CP110 removal, a negative regulator of ciliogenesis. The authors should at least discuss any potential mechanisms for CP110 removal by this SACAr, an important aspect.

The reduction of ADSL in human patients, as well as numerous experimental systems, was shown to elevate SAICAR/r and S-Amp/S-Ado levels (substrates of ADSL in their monophosphate and riboside forms, respectively). We have

focused on the riboside forms that are what is predominantly detected in patients and cell lines. We observed that upon ADSL depletion, ciliogenesis is impaired (after serum starvation). Nucleoside supplementation does not rescue the phenotype, indicating that a lack of purines does not underlie the phenotype. The administration of SAICAr phenocopies this, indicating that the accumulation of SAICAr, due to a disruption in DNPS specifically at the ADSL step, is therefore likely responsible for the effects on ciliogenesis observed following ADSL depletion. This is further corroborated by the fact that both MRT00252040 and MTX, that inhibit DNPS upstream of ADSL, can rescue the defect, while DNPS remains impaired. However, we cannot rule out that the physical depletion of ADSL per se leads to some unexpected signaling or effects on the stability of other proteins.

To address this further, we carried out an analysis of SAICAr levels following

ADSL depletion in RPE-1 cells. Unfortunately, these cells have very low levels of SAICAr and we are technically unable to detect it at levels where we can reliably assess differences (new Figure 1—figure supplement 2B-C). In experiments in HeLa cells performed in parallel, we can readily detect SAICAr and its increase following ADSL depletion – note that levels in HeLa are roughly 250X higher. We would be very surprised if a similar relative increase was not occurring in RPE-1 but due to differences in core metabolism, we are simply below the limits of reliable detection of differences in that cell line. While we still believe that SAICAr accumulation is the most likely explanation, we have recognized this caveat in the interpretation of the data and altered the text accordingly. In addition, we revised the title to Pathway, rather than Metabolite, specific effects, to take into account other possibilities.

How SAICAr would impair CP110 removal and ciliogenesis remains an important and interesting question that we have tried extensively to address. We have revised the discussion to highlight several possibilities but will mention a few here.

SAICAR has been shown to activate PKM2, that can then phosphorylate numerous other kinases in the context of cancer cells (Keller et al., 2014, 2012). There is not a clear connection between PKM2 and ciliogenesis that we are aware of. However, its effect on a particular kinase or other enzyme could influence the process indirectly through signal transduction.

One putative PKM2 target identified was AKT3. AKT signalling was recently implicated in controlling ciliogenesis upstream of CP110 removal. We considered that ADSL depletion and SAICAr administration could then potentially lead to increased AKT3 activation that would impair ciliogenesis in the absence of serum. However, we cannot detect any indication that AKT is activated by SAICAr treatment, suggesting that this is not the underlying mechanism. However, it remains possible that aberrant kinase activation by a yet unidentified kinase (or kinases) underlies the phenotype. In future work, it would be of interest to address this with phosphoproteomics but this is out of the scope of the current manuscript.

Alternatively, ADSL may interact with and regulate a protein involved in ciliogenesis. Proximity labeling using multiple centrosomal baits identified ADSL as an interactor, suggesting that ADSL (or a subset of the ADSL pool) may localize to the cilia(Gupta et al., 2015). We attempted to validate this by examining ADSL localization in IF, but could not detect a clear centrosomal signal. However, it remains possible that a subset of the ADSL interacts directly with centrosomal proteins to regulate their functions in some way.

We have extensively revised the discussion to include these possibilities and accompanying references.

The authors discuss p53 dependent cell death in Seckel syndrome (or when CPAP is eliminated). This is an important point that stem cell death causing microcephaly that is so far observed only in mouse models (few works that used CPAP elimination). It is somehow misinterpreted in the field that stem cell death is a significant cause of microcephaly. In the present work, this is not the case which is new evidence. It is also true that when Seckel syndromic microcephaly was modeled in human brain organoids (By Gabriel et al. modeling microcephaly due to CPAP mutation and WDR67 KO). Importantly, these experiments used disease-relevant cell types that did not show any cell death, but the microcephaly was due to cilia abnormality. It is disappointing why the authors ignore these essential works, which support their current work. Addressing these clarifications should result in a solid manuscript for publication.

We appreciate the reviewer pointing out these details, it is clearly relevant to discuss, as we indeed did not see apoptosis in vivo. Due to involvement in other work addressing apoptosis mechanisms, our discussion was admittedly limited to this area. As previously discussed, we did examine cilia removal after serum starvation and did not see defects in ADSL depleted cells. While we cannot discount that defects in cilia are related to the cell cycle issues we observed in chicken embryos, we think it is unlikely to be the main issue here based on this experiment. In the revision, we extended the discussion and add additional related references to explore this issue further.

Minor:It would be helpful if the authors explain what is SAICAr and S.Ado in the abstract itself. The abstract is very convoluted for a non-specific reader. I got the meaning of the abstract after reading the manuscript.

Based on several other comments, we have tried to make the abstract less technical in this regard and we added a schematic (new Figure 1—figure supplement 1) that shows the relationship between the enzymes and metabolites as suggested by another reviewer to address this point (see below). We also added a phenotype table (Appendix 1) to allow faster comparison of cellular and organismal phenotypes and the ID of the relevant figure panels.

Reviewer #1 (Significance (Required)):See above.Reviewer #2 (Evidence, reproducibility and clarity (Required)):The authors chose to investigate the link between ADSL deficiency and microcephaly by studying the impact of ADSL experimental depletion on cell proliferation, fitness and fate, using a combination of in vitro and in vivo approaches and different cell and model organisms, i.e. RPE1 cells, chicken and zebrafish. The authors propose that both nucleotide depletion and intermediate metabolite production are responsible for neurodevelopmental defects. Furthermore, they show impaired cell cycle progression, defective primary ciliogenesis, as well as DNA damage after ADSL depletion. The manuscript is of good quality, well written and referenced. The data and methods are clearly presented, illustrated and discussed. I have however two main criticisms regarding the manuscript.Major criticisms:My first criticism concerns the conclusions taken on the consequences of reduced purine levels.1. The contribution of reduced purine levels after ADSL depletion is only relying on the capacity of experimental nucleoside supplementation to rescue or not the phenotype of interest. Nonetheless, there are no clear information throughout the manuscript on the choice of dose used to supplement the purine deficiency and supposedly restore nucleoside concentration, which is to my sense a critical point. More importantly, we cannot appreciate whether such addition of nucleosides is readily compensating the deficiency without generating another imbalance. The authors should evaluate quantitatively whether their experimental conditions by nucleoside supplementation are sufficient to restore or at least to which extent they restore purine-based nucleoside concentration.

The nucleoside supplementation experiments were performed using a concentration based on numerous other experiments that used this intervention to address replication stress (Halliwell et al., 2020; Ruiz et al., 2015; Kanu et al., 2016; Parsels et al., 2018). This is a media supplement that has been widely used in the replication stress field and sufficient to rescue genetic and pharmacologically induced replication stress. Our results indicate that this is sufficient to functionally restore nucleotide levels, including after treatment with Hydroxyurea (new Figure 2E), however we recognize that they are most likely in excess of what they were prior to treatment.

We repeated key experiments with the addition of only Adenosine, as suggested by the reviewers (see new Figure 1—figure supplement 1) and we observed a similar rescue of specific phenotypes associated with ADSL depletion, indicating this is likely due to the restoration of limiting purine nucleotides, as we proposed. We have also added new data in zebrafish that shows that nucleoside supplementation of embryos is not sufficient to rescue the reduction in Sox2+ progenitor levels. This is consistent with data from human patients that have normal purine levels and the fact that we can partially rescue progenitors using MTX, that targets the pathway upstream of ADSL, thus implicating metabolic intermediates such as SAICAr rather than purine deficiency.

We do not feel it is feasible or necessary to repeat all experiments to measure precise levels of individual nucleosides/tides, as this is unlikely to make the data more informative and we do believe it will not be trivial to achieve consistent physiological levels in this type of cell culture addback experiment.

2. The effects of nucleoside supplementation can also be questioned after observing that nucleoside addition in control conditions appears sufficient to decrease the percent of Ki67 positive cells (Figure 1I), p53 positive cells (Figures 1J) and the percent of gH2Ax nuclear intensity (Figure 1D). The authors should explain this observation in the manuscript.

It is true that we observed minor alterations in the percentages of these parameters in control cells following nucleoside supplementation. In the case of Ki67, these effects are marginal (Figure 1I) and not statistically different. The fact that p53 and gH2AX are also slightly reduced (Figure 1J and 2D) is not entirely surprising and it is consistent with the fact that nucleosides may suppress culture stress arising from atmospheric oxygen. Similar observations have been made previously and this is why nucleoside supplementation is used in culture of ESCs and iPS cells.

Nucleosides of course have many other cellular roles and we do not see any possible way that it is feasible to ensure they are only used in DNA replication, regardless of how carefully the concentration is controlled. However, I think that the important point here is that the clear difference between control and ADSL depletion remains with regards to reduced Ki67 and increased p53 despite the minor effects on control RPE-1.

In contrast to p53 and Ki67, nucleosides suppress DNA damage signaling that we

believe is arising from mild replication stress. Treatment of cells with hydroxyurea (HU), a ribonucleotide reductase inhibitor, is a common intervention used to induce replication stress by limiting the nucleotide pool. We tested this nucleoside intervention on HU treated cells and found it was also sufficient to strongly reduce DNA damage (as has been previously published)(new Figure 2E). Therefore, we know that this intervention is capable of rescuing a strong inhibition of the ribonucleotide reductase.

As mentioned, we examined key endpoints with the restoration of only Adenosine to determine if this is phenotypically similar and obtained similar results, indicating that restoration of the purine pool is the most likely explanation.

My second main criticism regarding the manuscript is that the authors describe the different phenotypes that they observe, without exploring at all whether these defects are independent or related one to each other. This is an issue, as we can clearly expect from the literature that some phenotypes are related one to each other. Furthermore, some conclusions are not convincing, some phenotypes are not enough explored and some experiments are clearly lacking.

We agree that the phenotypes may be inter-related but we feel that we made a strong effort to determine which phenotypes result from reduced purine pools (ex. DNA damage/p53/cell cycle arrest) vs those that result from impaired DNPS and intermediate metabolite accumulation (ex. Impaired ciliogenesis). These have been substantiated with rescue experiments using nucleosides, SAICAr administration and small molecule inhibitors of the pathway upstream of ADSL.

Moreover, we attempted to determine how these impact particular aspects of the pathology observed in vivo by complementation experiments with nucleosides or inhibition of DNPS/SAICAR production where feasible. We clearly did not convey this as effectively as intended and to address this we have included a schematic model of the pathways (Figure 1-supplement 1), as well as a summary of phenotypes and interventions (Appendix 1) in the revised manuscript to try to clarify this point further. We hope this addresses the reviewer’s comments but without specific examples, we are not fully confident we understand the issue completely.

3. The authors mention that DNA damage signalling, evaluated after 53BP1 and gH2Ax immunostaining, is mildly affected, but can be suppressed by nucleosidesupplementation (Figure 2). It is difficult to reach this conclusion with the data presented in Figure 2. If this is a sound conclusion for 53BP1, only a very small population of cells (maybe 10% of cells) show differences in gH2Ax staining intensity between control and ADSL siRNA conditions (Figure 2D). One possibility is that DNA damage is repaired and only transiently detected during the cell cycle.

We have 2 issues at play in our opinion. The first is the use of siRNA- this leads to different levels of depletion in individual cells and ADSL was difficult to deplete, requiring 2 subsequent transfections of siRNA. Unfortunately, ADSL deletion is lethal and using a stable shRNA leads to growth defects prior to the experiment, making it difficult to compare to controls. It is likely that cells with the strongest depletion will not exhibit DNA damage due to the fact they exit cell cycle (replication is necessary to elicit replication stress). Likewise, cells with poor depletion will not suffer a significant reduction in purine levels. This would be consistent with the fact that we see a subpopulation of cells that exhibit DNA damage markers. A more continuous detection method would be ideal but is not feasible due to the siRNA depletion protocol needed to efficiently reduce ADSL.

We attempted to perform ADSL immunofluorescence, that would allow a single cell based analysis of depletion levels, but this signal is too weak and diffuse to use as a marker in these experiments to “normalize” the effects in any way. For this reason we provided population level graphing of gH2AX intensity or the presence of 53BP1 foci, as is standard in the field. It is clear that there is wide variability in the levels (which may also reflect cell cycle phase and timing of the damage related to DNA repair, as the reviewer points out, as well as the level of ADSL depletion in that particular cell).

The second issue is that the phenotype is mild. We provided IR treatment as a benchmark, given that this is what is commonly used to elicit damage markers. It is clear that there is a low amount of DNA damage compared to an exogenous insult like IR. That being said, we can reliably detect an increase in the intensity of both damage markers in the siADSL population that is reduced by nucleoside supplementation (notably this is also observed in vivo in zebrafish, further substantiating the phenotype – Figure 7G) and was recently reported following PAICS inhibition in leukemia cells. As cells are repair proficient, as the reviewer points out, this is potentially an underestimate of the phenotype due to repair of lesions.

In the revised manuscript we now provide additional example images in Figure 2, as well as an assessment of RPA levels in chromatin (Figure 2A: proxy measure of ssDNA) that accompanies replication stress to further address this point.

As cell cycle progression is also impaired after ADSL depletion (Figures 1D and 1G), the authors should evaluate at which cell cycle stage high levels of gH2Ax staining can be detected to explore this possibility.

As indicated in the response to reviewer 1, we carried out an additional analysis of the cell cycle using flow cytometry (new Figure 1—figure supplement 2E). In our experience, IF is a more sensitive method for quantitative gH2AX analysis and we repeated key experiments with the addition of Adenosine to address previous comments but were unfortunately not able to optimize coupling gH2AX with cell cycle analysis due to limitations arising from Covid restrictions and termination of fellowships. While this would be interesting, we remain unclear exactly what this would clarify. Similar to what is well documented with HU that depletes nucleotide pools, we expect that the primary issue is in S-phase (consistent with RPA results now presented) but damage can arise there due to fork stalling and breakage, in G2 due to condensation of under-replicated regions, in mitosis due to breakage of under-replicated regions and in G1, in the form of 53BP1 foci that mark those broken regions.

Furthermore, it is difficult to conclude about a rescue by nucleoside addition, as both control and ADSL siRNA conditions show decrease in gH2Ax intensity level. The authors need to explain this point in the manuscript. To strengthen their conclusions about DNA damage signalling, they have the possibility to evaluate the activation of classical signalling pathways that involve the kinases Chk1 and Chk2.

We respectfully disagree with the reviewer on this point. The fact that nucleoside addition reduces basal gH2AX levels in the WT cells is not entirely unexpected, as most cultured cells in atmospheric oxygen culture conditions have some level of replication stress that would be reduced by nucleoside addition. This nucleoside mix is sold commercially to address precisely this phenotype in normal stem cells.

The point we wish to make is that the clear difference between WT and ADSL depletion is now erased, indicating that the defect in ADSL depleted cells was likely due to insufficient nucleotide supply, rather than SAICAR accumulation, and this is consistent with new results using only Adenosine. Restoring nucleosides (likely in excess) erases the difference in damage markers between control and ADSL depleted lines, with the caveat that nucleosides may cause other off target effects.

Given how mild the phenotypes are, we do not expect that we will see a strong CHK1/2 response and have been unable to address this experimentally. We note that we did demonstrate that treatment with ATM inhibitors suppressed the 53BP1 increase, providing evidence that ATM signaling is active to some extent (Figure 2C).

4. From the literature, nucleotide depletion can trigger replication stress, and especially replication fork stalling in S phase as a source of DNA damage. This can trigger G2 block as observed in the chicken spinal cord (Figure 3D), due to DNA damage signalling activation before mitotic entry. Can the authors provide evidences of the likelihood of this scenario? Do they observe replication stress, or any sign of replication fork stalling? One possibility is to use RPA (Replication Protein A) loading as a readout of replication stress.

Replication stress can indeed induce a G2 arrest but this depends largely on the amount of damage being produced. For example, low dose aphidicolin induces clear levels of replication stress without eliciting a potent checkpoint response and under-replicated DNA enters mitosis where it can lead to strand breaks. This can also induce p53 and cause a G1 arrest, consistent with the data in RPE1 cells. As previously described, we now provide analysis of RPA, as well as cell cycle progression (Figure 2A and Figure 1—figure supplement 2E), that indicate that we have mild replication stress that is rescued by nucleosides/Adenosine and a G1 arrest that is p53 dependent.

5. Decreased % of ciliated cells and decrease in cilia length can be observed before mitosis in normal conditions, as cilium undocking from the plasma membrane and resorption occurs in general before mitotic entry. Can the authors exclude the possibility that the "defects" in primary ciliogenesis are not only related to the block in G2 phase that they observe (Figure 3F)? Further, CP110 is normally degraded in G2 phase by an F box protein. Any block in G2 phase would therefore be at the origin of CP110 accumulation on both centrioles. Can the authors explore this possibility?

This is a good point and one that we previously considered. First, we have not observed a block in G2 in RPE-1 cells. Unlike the NPCs in vivo, RPE-1 arrest in G1 in a p53-dependent manner and this is now confirmed by the cell cycle analysis presented in Figure 1—figure supplement 2E. As mentioned in the response to reviewer 1, we examined cilia removal in detail following serum addition post ciliation (new Figure 4—figure supplement 1G). This did not indicate any defects in the cilia removal process that is cell cycle dependent as the reviewer indicated. Additionally, we know that the cells being analyzed have left the cell cycle efficiently by loss of Ki67 staining (Figure 4A), so there is unlikely to be a stalled G2 population among the cells being analyzed for ciliogenesis. We therefore think that the cell cycle arrest observed in cycling RPE-1s does not account for the ciliogenesis defect.

Minor criticisms:1. Ki67 immunofluorescent staining panels provided in Figure 1H and 1L are not accurately illustrating differences in the percent of Ki67 positive cells between the different conditions.

We have tried to improve these images and indicate clearly what is being scored as positive vs negative cells (keep in mind that the DAPI shows all the cells and one must go back and forth a bit). We simply cannot get enough cells in the field to provide a single image that is mathematically representative of the total scoring in each case but believe the current images are highly representative of the populations observed. We are open to suggestions for better ways to annotate this.

2. 53BP1 immunofluorescent staining, which appears as very small dots, cannot be appreciated from the panels provided in Figure 2C.

We have again tried to improve the 53BP1 example images in what is now Figure 2B and 2D, including some enlarged cells so that examples of foice can be seen clearly. We hope this addresses the reviewer’s concern.

3. The dose used for nucleoside supplementation should be mentioned in the method section.

We agree and apologize for this omission, it has been corrected, thanks for pointing this out.

4. The title of the manuscript contains too many abbreviations and does not give a clear idea of the main findings of the manuscript. I recommend to change it.

We have considered this point, as well as other related comments from reviewers and changed the title to: Pathway specific effects of ADSL deficiency on neurodevelopment

Reviewer #2 (Significance (Required)):The manuscript by Dutto et al. aims to provide a better understanding of the underlying mechanisms that lead to the microcephaly phenotype that can be observed in some human patients affected by ADSL (Adenylosuccinate lyase) deficiency. As ADSL is an enzyme involved in the de novo purine synthesis, patients suffer from global defects in purine metabolism, but the link with neurological symptoms, and especially microcephaly when present, remains totally unexplored. First, the authors provide a characterization of the impact of ADSL depletion on cell proliferation, fitness, fate and outcome and propose that the consequences are context-dependent. Second, as the accumulation of intermediate metabolites like SAICAr has already been proposed as neurotoxic, they clarify the respective implication of purine depletion and SAICAr accumulation on the different cell phenotypes observed and on neurodevelopment. Beyond the conceptual advance provided by this work for the etiology of microcephaly, and the ADSL deficiency disorder in particular, the manuscript will be of interest for the developmental and cell biology communities regarding the impact of metabolism impairment at the cell and organism level.I am not personally an expert in cell metabolism, but in development and cell cycle progression.Reviewer #3 (Evidence, reproducibility and clarity (Required)):The evidence presented of the ADSL deficiency effect on cell proliferation, through DNA damage increase and p53 activation is highly compelling, as is the effect on ciliogenesis. The interpretation and conclusions however, in our view, raise two major points, that need to be addressed. The good news is that these don't necessary require new experiments.Major points:The intermediate products in the purine biosynthesis pathway are metabolized in the monophosphate form, particularly the ADSL substrates SAICAR (a.k.a. SZMP) and SAMP. To the best of our knowledge the monophosphate forms are the biologically active and physiologically relevant forms. Throughout the manuscript the observed results are ascribed to the riboside forms SAICAr and sAdo – including in the title of the manuscript – which are their degradation products, and the measurements of the monophosphate forms not mentioned and not presented. Given the metabolic profiles were performed using LC-MS this information will be probably available to the authors by re-analyzing their data, and presumably will not require further experiments. This issue needs to be addressed throughout the manuscript, the data presented does not grant ascribing the phenotypes to the riboside forms without establishing the status of the monophosphate forms.

We agree with this point with some caveats. The degradation (riboside) products are more readily detected in ADSLD patient serum and cell lines and specifically SAICAr has been implicated in pathological outcomes experimentally(Stone et al., 1998), however there is no data we know of indicating whether either form is more relevant for any particular phenotype in vertebrates.

We specifically used SAICAr for the cilia experiments and see that it phenocopies ADSL depletion, although we did not test SAICAR in parallel, and thus do not know if there is a substantial difference in their effect. As SAICAr accumulates in human patients and was specifically implicated in neuropathology(Stone et al., 1998) in mammals, we chose to examine it in isolation. We will revise the text to ensure we are not making conclusions that do not take this relevant point into consideration.

We attempted many times to measure the SAICAR/r levels without success, although reduced AMP, GMP and increased S-Ado were readily detected. We revisited this again with the help of the Zikanova lab and again carried out MS on RPE-1, also examining HeLa cells in parallel, where ADSL loss has been shown experimentally to induce SAICAR/r. While we can readily detect the expected modulation of SAICAr (and other metabolites) in HeLa cells, we simply cannot detect SAICAr in RPE1 to a level that is sufficient for meaningful comparisons – it is ~250X lower in RPE1 than HeLa. We have tried several optimization steps but his is a technical limitation we are unfortunately not able to overcome. This data does however give us some pertinent information regarding the fact that the PAICS inhibitor has the expected impacts on its accumulation using HeLa cells. A similar issue in detection was encountered previously in human fibroblasts during the characterization of a PAICS mutation and there is unfortunately not anything further we can do with currently available methods. Given this technical limitation, we have revised the conclusions to ensure we are not making an absolute cause-effect conclusion and accounting for the possibility that specifically depleting ADSL could have other effects not directly related to the accumulation of substrates.

The ADSL activity in converting SAMP into AMP is common to both de novo synthesis of AMP and AMP synthesis through the recycling pathway. This is relevant particularly when discussing the potential effects of sAdo/SAMP accumulation, which may be due to recycling rather than de novo synthesis. This point needs to be clarified throughout the manuscript.

There are many discrepancies in the literature related to the definition of the different pathways. To define this in our opinion, we included a schematic in new Figure 1—figure supplement 1 that breaks down the roles of ADSL in DNPS and the purine nucleotide cycle. Both DNPS and recycling pathways feed into the purine nucleotide cycle as the reviewer alludes to. We have gone through the text and revised our discussion of these pathways regarding the interpretation of the data but would ask for specific passages that the reviewer feels are incorrect if any remain. We have made very few conclusions regarding the potential effects of SAMP/S-Ado accumulation and have not used that to attempt to induce phenotypes, so the observation that this increases upon ADSL depletion is simply a confirmation to us that we are effectively impairing its activity.

Specific points (that require new experiments):In a number of instances the observed effects are ascribed to changes in the abundance of specific metabolites, however no evidence is provided that such changes occur (as measured by LC-MS).

We agree that it is important to try and confirm the effects of several interventions used in the paper to make key conclusions. We address the individual points below.

These are specifically: reduction of SAICAr/SAICAR levels with MRT00252040 drug (e.g. lines 122-130);

Our data indicates the drug is working specifically as expected in HeLa but due to technical limitations, we cannot determine this clearly for RPE1 (new Figure 1figure supplement 2B-C). We expect that a similar proportional effect is taking place in RPE1 but due to the much lower levels, we cannot make any quantitative statements and unfortunately do not have any additional methods by which to address this. All of the experiments using MRT00252040 have been further corroborated by the addition of SAICAr that leads to the same outcome of ADSL depletion. Given that we cannot definititely address changes in SAICAr levels in RPE-1, we have adjusted the text to reflect other possibilities.

Increase in SAICAr/SAICAR levels upon SAICAr treatment (Figure 4);

We are unclear what this would clarify. Given that we have a clear phenotypic effect, what does the LC-MS measurement of this added in excess inform us about and why would we measure SAICAR when we are adding SAICAr? Given the cost of this reagent and scale we require for MS, we cannot consider this experiment without a clearer rationale.

Increased levels of AMP and GMP upon treatment with nucleosides throughout the manuscript;

As discussed earlier, we are using a well-established intervention and we have experimentally validated that this is sufficient to suppress damage induced by HU (new Figure 2E). We have not seen anyone ever report nucleotide levels in the context of this type of experiment of adding nucleosides. We remain unclear again how quantifying this is essential given the number of phenotypic endpoints we provide supporting the fact that purine pools are restored, including adenosine add back that is now included in the revised manuscript.

Levels of SAICAr/SAICAR upon methotrexate treatment (e.g. Figure S3). These measurements are particularly important in the instances where negative results occur, compounds added in the external medium don't necessarily enter all cells or reach all tissues.

As mentioned, we are not able to meaningfully measure SAICAr levels in RPE1 due to detection limits. We also do not agree that measuring metabolite levels will address whether all cells or tissues have incorporated the drug (they surely have not). MTX has been used for decades and the effects are well documented, including that it significantly impacts AMP and GMP levels in cell culture at similar concentrations to what we used (Hoxhaj et al., 2017). While we cannot determine that MTX is inducing SAICAr due to the much lower levels in RPE-1, there is little doubt this is being incorporated and acting as expected in our opinion. In zebrafish, we are certain it is not being taken up by all cells/tissues, there is again no meaningful way to address this experimentally. However, the results of its addition are clear and reproducible in fish depleted for ADSL.

The authors use a mix of nucleosides to "rescue" ADSL deficiency, we would suggest to add an experimental group with Adenosine only in the same experimental conditions, as Adenosine alone should restore AMP (and even GMP levels). This experiment is not absolutely required, but would be interesting, we would leave it to the consideration of the authors.

We thank the reviewer for this great suggestion which was carried out. Adenosine rescues the same phenotypes of the nucleoside mix, indicating that, while there may be some off target effects, the main effect of these interventions is to restore the purine pool.

Suggestions for text:We would suggest a schematics of the purine biosynthesis pathway, placing the two reactions catalyzed by ADSL, on Figure 1. It's very helpful, even for specialists.

We agree and have added this as indicated previously in Figure 1—figure supplement 1 of the revised manuscript, as described previously in this document.

There is an ambiguity in the literature concerning some purine pathway intermediates including ADSL substrates. SAICAr – riboside form – is also referred in the literature as SAICAR, and SAICAR – monophosphate form – is also referred to SZMP. We would suggest, to avoid ambiguity" to mention once in the text "SAICAR (monophosphate from a.k.a. SZMP)".

We added this for clarification in the text.

State once in the text the which nucleosides compose the nucleoside mix, and in the Materials and methods detail the composition and concentrations of each nucleoside.

This information was partially included in the methods section, however we agree that the precise details are important. We updated this with additional details in the revised manuscript. We used a commercial mix (Embryomax) that has been used previously in numerous papers for attenuating replication stress. This contains Cytidine (0.73 g/L), Guanosine (0.85 g/L), Uridine (0.73 g/L), Adenosine (0.8 g/L) and Thymidine (0.24 g/L)(Halliwell et al., 2020; Ruiz et al., 2015; Kanu et al., 2016; Parsels et al., 2018).

Line 547 "morpholino targeting" should read "morpholino targeting Adsl".

We have corrected this, thank you.

Statistics and experimental design:We would suggest consult with a statistician and revise the manuscript accordingly.Here are the issues we detected.– t-test not suitable for experiments with n=4, better use a non-parametric test (e.g. metabolic profiling Figure 1C).

We have consulted our statistician and we respectfully disagree with the reviewer on this point. The adequacy of a test depends on the distribution generating the data and not on the number of observations. When a distribution cannot be assumed given the nature of the experiment or normality cannot be tested, a nonparametric distribution should be used. Nevertheless, for Figure 1C and similar experiments which include continuous positive measurements we decided to apply a logarithmic transformation to ensure normality. Revised statistical analysis is presented in the revised manuscript.

– For normalisation of western blots, total protein is more suitable than actin.

We included another representative blot with a corresponding Ponceau panel and included this in new Figure 1B.

– Percentages and n (e.g. Ki67 or p53 positive cells) is ambiguous, we would suggest to mention both how many replicates were performed (which as we understand correspond to the stated n) and how many cells were scored (could be an average per replicate).

We prefer to graph the percentages, as we are combining replicates of independent experiments. We revised legends to indicate clearly the n, as well as the cell numbers where applicable in the revised manuscript.

– Anova and t-student test are suitable for continuous variables, not for percentages (averaging percentages does not make the variable continuous), contingency tables and Fisher's exact test, or statistical tools based on the same principle, are more suitable.

We applied a generalized linear model with a binomial distribution in order to capture the nature of the positive/negative values of the cells in the experiment, and to take into consideration multiple biological replicates. Resulting coefficients, p-values and methods are included in the updated manuscript.

– Whenever multiple comparisons are performed, a Bonferroni correction should be applied. If that was included in the statistical analysis it should be stated.

We thank the referee for this suggestion but do not feel it is necessary to apply a correction in most experiments. Each of our experiments measure the percentage of positive cells under different and independent conditions. Moreover, the hypotheses of interest were decided before generating the data and all associated p-values are reported regardless of their significance. We believe that conceptually this discards the need for adjusting for multiple hypothesis testing.

Also, for those cases where multiple contrasts are extracted from the same linear model and dataset, we have applied the Shaffer correction method as implemented in the "multcomp" R package.

– Where the experiments performed blindly? If so it should be stated. If not we would suggest for future work that treatment/control groups are blinded for the observer scoring the results.

Experiments were scored with the samples blinded in all cases. We will indicate this in the methods section as requested.

Reviewer #3 (Significance (Required)):The manuscript RC-2021-00642 by Dutto et al. presents a characterization of ADSL deficiency in three different model systems. It provides compelling evidence of ADSL deficiency causing proliferation defects, through p53 activation, and in cilliogenesis. The defects observed particularly in the nervous system provide indications of the etiology of patients affected by ADSL deficiency. Hence this research provides new and relevant information concerning the understanding not only of ADSL deficiency but other genetic diseases linked to defects in the purine biosynthesis pathway. Beyond the medical aspect this article provides rather novel insights onto the impact of metabolism, and in particular biosynthetic pathways, have in developmental processes. There is very little research on this topic on the literature, particularly in the developmental models reported: zebrafish and chick embryo. Overall the paper is of interest both to the field of genetic metabolic diseases and developmental biology.

References:

Gabriel, E., A. Wason, A. Ramani, L.M. Gooi, P. Keller, A. Pozniakovsky, I. Poser, F. Noack, N.S. Telugu, F. Calegari, T. Šarić, J. Hescheler, A.A. Hyman, M. Gottardo, G. Callaini, F.S. Alkuraya, and J. Gopalakrishnan. 2016. CPAP promotes timely cilium disassembly to maintain neural progenitor pool. EMBO J. 35:803–819. doi:10.15252/embj.201593679.

Gupta, G.D., É. Coyaud, J. Gonçalves, B.A. Mojarad, Y. Liu, Q. Wu, L. Gheiratmand, D.

Comartin, J.M. Tkach, S.W.T. Cheung, M. Bashkurov, M. Hasegan, J.D. Knight, Z.-Y. Lin, M. Schueler, F. Hildebrandt, J. Moffat, A.-C. Gingras, B. Raught, and L. Pelletier. 2015. A Dynamic Protein Interaction Landscape of the Human Centrosome-Cilium Interface. Cell. 163:1484–1499.

doi:10.1016/j.cell.2015.10.065.

Halliwell, J.A., T.J.R. Frith, O. Laing, C.J. Price, O.J. Bower, D. Stavish, P.J. Gokhale, Z. Hewitt, S.F. El-Khamisy, I. Barbaric, and P.W. Andrews. 2020. Nucleosides Rescue Replication-Mediated Genome Instability of Human Pluripotent Stem Cells. Stem Cell Rep. 14:1009–1017. doi:10.1016/j.stemcr.2020.04.004.

Hoxhaj, G., J. Hughes-Hallett, R.C. Timson, E. Ilagan, M. Yuan, J.M. Asara, I. BenSahra, and B.D. Manning. 2017. The mTORC1 Signaling Network Senses Changes in Cellular Purine Nucleotide Levels. Cell Rep. 21:1331–1346.

doi:10.1016/j.celrep.2017.10.029.

Kanu, N., M.A. Cerone, G. Goh, L.-P. Zalmas, J. Bartkova, M. Dietzen, N. McGranahan,

R. Rogers, E.K. Law, I. Gromova, M. Kschischo, M.I. Walton, O.W. Rossanese, J. Bartek, R.S. Harris, S. Venkatesan, and C. Swanton. 2016. DNA replication stress mediates APOBEC3 family mutagenesis in breast cancer. Genome Biol. 17:185. doi:10.1186/s13059-016-1042-9.

Keller, K.E., Z.M. Doctor, Z.W. Dwyer, and Y.-S. Lee. 2014. SAICAR induces protein kinase activity of PKM2 that is necessary for sustained proliferative signaling of cancer cells. Mol. Cell. 53:700–709. doi:10.1016/j.molcel.2014.02.015.

Keller, K.E., I.S. Tan, and Y.-S. Lee. 2012. SAICAR stimulates pyruvate kinase isoform M2 and promotes cancer cell survival in glucose-limited conditions. Science.

338:1069–1072. doi:10.1126/science.1224409.

Parsels, L.A., J.D. Parsels, D.M. Tanska, J. Maybaum, T.S. Lawrence, and M.A. Morgan. 2018. The contribution of DNA replication stress marked by highintensity, pan-nuclear γH2AX staining to chemosensitization by CHK1 and WEE1 inhibitors. Cell Cycle. 17:1076–1086. doi:10.1080/15384101.2018.1475827.

Piazza, I., K. Kochanowski, V. Cappelletti, T. Fuhrer, E. Noor, U. Sauer, and P. Picotti. 2018. A Map of Protein-Metabolite Interactions Reveals Principles of Chemical Communication. Cell. 172:358–372.e23. doi:10.1016/j.cell.2017.12.006.

Ruiz, S., A.J. Lopez-Contreras, M. Gabut, R.M. Marion, P. Gutierrez-Martinez, S. Bua,

O. Ramirez, I. Olalde, S. Rodrigo-Perez, H. Li, T. Marques-Bonet, M. Serrano, M.A. Blasco, N.N. Batada, and O. Fernandez-Capetillo. 2015. Limiting replication stress during somatic cell reprogramming reduces genomic instability in induced pluripotent stem cells. Nat. Commun. 6:8036. doi:10.1038/ncomms9036.

Stone, T.W., L.A. Roberts, B.J. Morris, P.A. Jones, H.A. Ogilvy, W.M. Behan, J.A.

Duley, H.A. Simmonds, M.F. Vincent, and G. van den Berghe. 1998.

Succinylpurines induce neuronal damage in the rat brain. Adv. Exp. Med. Biol. 431:185–189. doi:10.1007/978-1-4615-5381-6_36.